# OPHR: Mastering Volatility Trading with Multi-Agent Deep Reinforcement Learning

**Zeting Chen**[*]
Nanyang Technological University
Singapore
zchen091@e.ntu.edu.sg

**Xinyu Cai**[*]
Nanyang Technological University
Singapore
xinyu009@e.ntu.edu.sg

**Molei Qin**
Nanyang Technological University
Singapore
molei001@e.ntu.edu.sg

**Bo An**
Nanyang Technological University
Skywork AI
Singapore
boan@ntu.edu.sg

## Abstract

Options markets represent one of the most sophisticated segments of the financial ecosystem, with prices that directly reflect market uncertainty. In this paper, we introduce the first reinforcement learning (RL) framework specifically designed for volatility trading through options, focusing on profit from the difference between implied volatility and realized volatility. Our multi-agent architecture consists of an Option Position Agent (OP-Agent) responsible for volatility timing by controlling long/short volatility positions, and a Hedger Routing Agent (HR-Agent) that manages risk and maximizes path-dependent profits by selecting optimal hedging strategies with different risk preferences. Evaluating our approach using cryptocurrency options data from 2021-2024, we demonstrate superior performance on BTC and ETH, significantly outperforming traditional strategies and machine learning baselines across all profit and risk-adjusted metrics while exhibiting sophisticated trading behavior. The code framework and sample data of this paper have been released on https://github.com/Edwicn/OPHR-MasteringVolatilityTradingwithMultiAgentDeepReinforcementLearning

## 1 Introduction

Options markets represent one of the most sophisticated segments of the financial ecosystem, offering traders the ability to construct complex, non-linear payoffs and express nuanced views on market direction, volatility, and timing. More recently, cryptocurrency options markets have emerged as a rapidly developing segment, with Bitcoin (BTC) [Nakamoto, 2008] and Ethereum (ETH) [Buterin and Others, 2013] options gaining significant traction, which provide unique research opportunities due to their transparency, 24/7 operation, and comprehensive data accessibility that is often unavailable in traditional markets.

Options [Kariya and Liu, 2003] serve as the insurance of financial markets, with prices that directly reflect market uncertainty. While traditional option pricing models like Black-Scholes-Merton (BSM) [Black and Scholes, 1973, Merton, 1973] provide a theoretical framework for option valuation, they rely on idealized assumptions that rarely hold in practice: continuous trading, no transaction costs, constant volatility, and the ability to hedge risk exposures perfectly. These simplifications,

---

[*]Equal contribution. Correspondence to: Xinyu Cai - xinyu009@e.ntu.edu.sg

39th Conference on Neural Information Processing Systems (NeurIPS 2025).

while mathematically elegant, fail to capture the empirical realities of options markets, where the difference between implied volatility (IV) and realized volatility (RV) creates persistent volatility trading opportunities [Carr and Madan, 1998, Christensen and Prabhala, 1998, Ni et al., 2008, Sinclair, 2013, Tan et al., 2024].

IV is the volatility priced in the option price [Canina and Figlewski, 1993, Dumas et al., 1998], and RV is the true price fluctuations in the underlying asset[Andersen et al., 2003, McAleer and Medeiros, 2008]. IV is higher than RV most of the time because the market needs to pay a premium to option sellers as a reward for bearing tail risk. Option sellers typically need to trade the underlying asset to hedge risk. Options aren't just insurance; during periods of significant market volatility, traders can often profit by buying options and properly hedging. In these instances, RV typically exceeds IV. Therefore, the volatility trading involves two challenges to beat the market: **i) Good volatility timing:** sell options to collect premium when markets are calm, and buy options to profit when markets are about to become volatile; **ii) Select a proper hedging strategy** to manage risk when selling options and take profit when buying options as illustrated in Figure 1.

RL [Sutton and Barto, 1998, Mnih et al., 2015, Hasselt et al., 2016] offers a compelling alternative by learning optimal trading policies directly from market data. Unlike traditional approaches relying on option pricing models with restrictive assumptions and RV forecasting [Andersen et al., 2003], RL frames option trading as a sequential decision problem, naturally incorporating practical constraints such as transaction costs, discrete hedging intervals, and market dynamics. This data-driven methodology eliminates the need for explicit pricing models, instead allowing RL agents to discover mispriced volatility through experience.

Although RL has been successfully applied to many trading tasks [Jiang and Liang, 2017, Conegundes and Pereira, 2020, Yang et al., 2020, Briola et al., 2021, Nagy et al., 2023, Takara et al., 2023, Zong et al., 2024, Qin et al., 2024], options trading is an untouched area due to its complexity. In this paper, we introduce the first RL framework that addresses the complexities of volatility trading in a data-driven manner. Our framework is composed of 2 parts: **i) Option Position Agent (OP-Agent)** controlling the long/short of volatility, and **ii) Hedger Routing Agent (HR-Agent)** selecting the optimal **Hedgers** with different risk preferences to perform dynamic Delta hedging based on positions and market conditions. To make they work coordinately with each other, we first distill a sub-optimal Oracle policy to OP-Agent, then alternatively train HR-Agent and OP-Agent.

We evaluate our approach using historical cryptocurrency options data on BTC and ETH, leveraging the transparency and data accessibility of these markets to conduct comprehensive experiments. Our findings demonstrate that the RL model significantly outperforms traditional rules-based strategies in both long and short Gamma implementations, with particular emphasis on managing tail risk when selling options and identifying optimal timing for buying options during market dislocations.

The primary contributions of our work are:

- The first RL framework specifically designed for trading volatility through options, moving beyond previous RL applications that focused solely on option pricing and hedging.

- A novel multi-agent architecture comprised of two specialized agents working in concert: OP-Agent, identifying volatility trading opportunities, and HR-Agent, tuning risk preference.

- An effective training methodology that enables the two agents to learn collaboratively from market data, balancing the competing objectives of profit maximization and risk management.

- Comprehensive empirical results demonstrating the framework's ability to properly manage tail risk when selling options and identify advantageous timing for buying options.

## 2   Background & Related Works

This section introduces the fundamental concepts of options, option pricing models, and volatility trading strategies that form the basis for our RL approach to trade volatility.

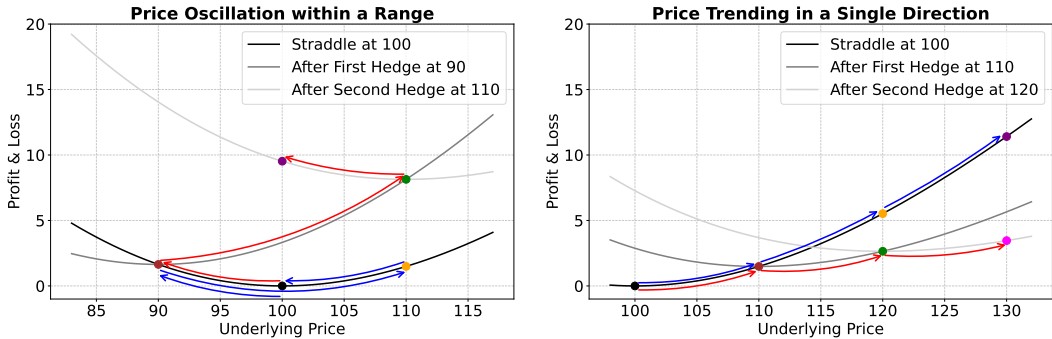

Figure 1: **Importance of proper Hedging strategy:** In this figure, red arrows show the PnL path with hedging, and blue arrows show the PnL path without hedging. In the oscillation market (left), the price path is $100 \rightarrow 90 \rightarrow 110 \rightarrow 100$. In the trending market (right), the price path is $100 \rightarrow 110 \rightarrow 120 \rightarrow 130$. These figures ignore the time value decay for simplicity. All curves in the two figures will drop as time passes.

## 2.1 Options and the Black-Scholes Model

Financial options are derivative contracts that give the holder the right, but not the obligation, to buy (call option) or sell (put option) an underlying asset at a specified price (strike price) on or before a specified date (expiration date). Options are characterized by three key attributes: **option type** (Call or Put), **strike price** ($K$) at which the underlying asset may be transacted, and **expiry time** ($T$) by which the option must be exercised. The payoff structure of European options is defined mathematically as:

$$\Psi(S_T) = \begin{cases} (S_T - K)_+ & \text{if Call} \\ (K - S_T)_+ & \text{if Put} \end{cases} \tag{1}$$

where $S_T$ is the underlying asset's price at expiration and $(x)_+$ mean $\max(x, 0)$.

The **Black-Scholes-Merton Model (BSM)** represents the foundational framework for pricing options. While real markets violate several of its assumptions, the model provides crucial theoretical insights and practical tools for options traders. The BSM assumes a constant risk-free interest rate $r$, an underlying asset price following geometric Brownian motion ($dS_t = \mu S_t dt + \sigma S_t dW_t$), frictionless markets with no transaction costs, and no arbitrage opportunities. These assumptions lead to the Black-Scholes partial differential equation: $\frac{\partial V}{\partial t} + \frac{1}{2}\sigma^2 S^2 \frac{\partial^2 V}{\partial S^2} + rS\frac{\partial V}{\partial S} - rV = 0$. Solving this equation yields the theoretical price of a European option $V(t, S_t)$.

Beyond pricing, BSM provides essential risk exposure metrics known as "**Greeks**". **Delta** ($\Delta = \frac{\partial V}{\partial S}$) quantifies an option's price sensitivity to changes in the underlying asset price. **Gamma** ($\Gamma = \frac{\partial^2 V}{\partial S^2}$) measures the rate of change in $\Delta$ relative to movements in the underlying asset. **Theta** ($\Theta = \frac{\partial V}{\partial t}$) measures time decay—how much value an option loses as time passes. **Vega** ($\frac{\partial V}{\partial \sigma}$) quantifies sensitivity to volatility changes. These Greeks provide critical insights for risk management and volatility trading strategies: traders can hedge those risks they do not want and keep those they are willing to take. The details of BSM and Greeks are presented in Appendix A.1

## 2.2 Volatility Trading

There are many volatility trading strategies due to the complexity of options. One of them is the long/short Gamma strategy, which aims to profit from the difference between IV—the volatility priced into options by the market, and RV—the actual price fluctuations experienced by the underlying asset.

**The Gamma-Theta Relationship.** Options positions with positive $\Gamma$ experience convex payoffs (accelerating gains) as the underlying price moves in either direction. However, this advantage comes at a cost: negative $\Theta$, or time decay. This fundamental relationship informs the core mechanics of volatility trading. Using Taylor's theorem, we can approximate how an option's price changes:

$$V(S + dS, \sigma, t + dt) \approx V(S, \sigma, t) + dS \times \Delta + \frac{1}{2}(dS)^2 \times \Gamma + dt \times \Theta \tag{2}$$

When Delta-hedging is applied (neutralizing the $dS \times \Delta$ term through offsetting positions in the underlying), the theoretical PnL becomes:

$$\text{PnL} = \int_0^T \frac{S_t^2}{2} \Gamma \left( \sigma_t^2 - \sigma^2 \right) dt, \tag{3}$$

where $\sigma_t = (\frac{dS_t}{S_t})^2$ is the instant RV (**profit**) and $\sigma$ is IV which determines the magnitude of the time decay rate $\Theta$ (**loss**). The formula shows that if the expectation of RV($\sigma_t$) is higher than the IV($\sigma$) priced in options, we are expected to profit from a long Gamma trade and vice versa.

**Dynamic Delta Hedging for Gamma Strategies.** Dynamic Delta hedging (DDH) [Hull and White, 2017] serves distinct functions in the Gamma strategy. For long Gamma positions, it acts as a profit mechanism through $\Gamma$ scalping—systematically buying low and selling high as price fluctuations alter $\Delta$ values. For short $\Gamma$ positions, it primarily mitigates risk by neutralizing directional exposure. As illustrated in Figure 1, the effectiveness of hedging depends on market conditions. In oscillating markets, hedging locks in profits when prices return to original levels, while in trending markets, hedging may limit potential gains as prices move consistently in one direction. This demonstrates why proper hedging strategy selection is crucial despite similar RV in both scenarios. Rule-based Hedger is commonly used, which monitors portfolio Delta against predefined thresholds. Alternatively, RL approaches can be employed to train hedging strategies with varying risk-aversion levels through actor-critic methods using risk-adjusted reward functions [Buehler et al., 2019, 2021a,b, 2022, Murray et al., 2022]. While recent work such as DLOT [Tan et al., 2024] applies deep learning to option portfolio management by constructing long/short positions across multiple stocks' straddles without underlying hedging, our work addresses a fundamentally different problem: single-asset volatility trading with dynamic delta hedging. This distinction is crucial—our approach tackles the complex challenges of gamma scalping, continuous hedging decisions, and path-dependent profit optimization that emerge from maintaining isolated volatility exposure, requiring delta-hedged straddle PnL as the optimization target rather than unhedged portfolio returns.

## 3 Problem Formulation

We first present several financial concepts that are necessary to trade options. Then we describe how to formulate the volatility trading problem as a Cooperative Markov Decision Process (MDP).

**Financial Concepts for Volatility Trading.** To build an RL framework for volatility trading, we first define the following financial concepts: **Underlying asset**, denoted as $S_t$, refers to the financial asset from which the options are derived. In our context, it primarily includes the perpetual future of the underlying asset, which is used to hedge the $\Delta$ exposure of our volatility trading positions. **Straddle** is a combination of a call and a put option with identical strike prices. At the money (ATM, strike $K$ equals $S_t$) straddle is a common strategy to trade volatility due to its zero $\Delta$ exposure and substantial $\Gamma$ and Vega exposure. **Position**, denoted as $P_t$, consists of option positions: the holding of option contracts, and hedging positions: the holding of the underlying asset. **Net value** $V_t$ represents the total value of our account, which consists of the options positions value, underlying position value, and cash: $V_t = V_t^{options} + V_t^{underlying} + CashBalance$. **Market features** $F_t$ are features derived from options and underlying market data to analyze price trends and volatility. **Greeks**, $G_t = (\Delta, \Gamma, \Theta, \text{Vega})$, are risk measures derived from the BSM model, indicating the sensitivity of the net portfolio value ($V_t$) to different variables, which have been described in Section 2.1. **Hedgers** are hedging policies that take information like Greeks and time-to-expires as input, and decide how many hedging instruments to trade at each step, which have been described in Appendix B.2.

**Markov Decision Process Formulation.** We formulate volatility trading as a Cooperative MDP, where two specialized agents collaborate to optimize trading performance. The OP-Agent identifies and executes volatility trading opportunities, while the HR-Agent selects optimal hedging strategies every $N$ steps. This cooperative relationship is critical: the OP-Agent's position decisions directly influence the HR-Agent's state space, while the HR-Agent's hedging strategy affects the returns of positions established by the OP-Agent. Both agents share the common objective of maximizing portfolio net value while managing risk exposure. Specifically, in our framework, the problem can be formulated as $(MDP^{\text{op}}, MDP_N^{\text{hr}})$

$$
\begin{aligned}
MDP^{\text{op}} &= <S^{\text{op}}, A^{\text{op}}, T^{\text{op}}, R^{\text{op}}, \gamma^{\text{op}}> \\
MDP_N^{\text{hr}} &= <S_N^{\text{hr}}, A_N^{\text{hr}}, T_N^{\text{hr}}, R_N^{\text{hr}}, \gamma^{\text{hr}}>
\end{aligned} \tag{4}
$$

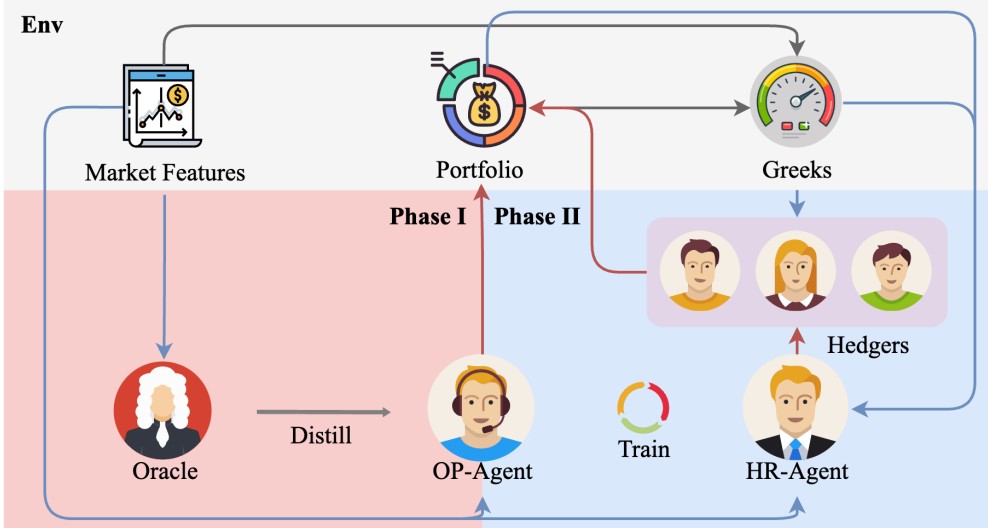

Figure 2: **The overview of OPHR.** The upper section depicts the trading environment containing market features, portfolio status, and Greeks measurements: both market and portfolio will affect the Greeks. The lower section shows the two-phase training process: In Phase I, a sub-optimal Oracle policy distills market knowledge to initialize the OP-Agent. In Phase II, the OP-Agent is trained alternatively with the HR-Agent, who selects a proper Hedger based on Greeks and market features.

**OP State** is the market feature $F_t$ , denoted as $s_t^{\text{op}} = F_t$. **OP Action**, denoted as $a_t^{\text{op}} \in A^{\text{op}} = \{+1, -1, 0\}$, determines the target position (long/short/neutral) of options at each step. **HR State**, denoted as $s_t^{\text{hr}} = (F_t, P_t, G_t)$, consists of the market feature $F_t$, position information $P_t$, and Greeks $G_t$. **Hedgers** are a set of hedging policies $\{\pi_{(i)}^{\text{hedger}}\}_{i=1}^{K}$ with different risk-aversion levels, which the method proposed by Murray et al. [2022] trains . A rule-based Hedger $\hat{\pi}^{\text{hedger}}$ is used as a baseline in our framework. They take the portfolio and Greeks $(P_t, G_t)$ as input and output the final trading amount of the underlying asset to perform a hedge. **HR Action**, denoted as $a_t^{\text{hr}} \in A^{\text{hr}}$, involves selecting the optimal Hedger from the set of predefined Hedgers In Figure 1, the two Hedgers are price-based Hedgers. The red one has a finite threshold (less risk-seeking), and the blue one has an infinite threshold (never hedging, extreme risk-seeking). Details of Hedgers are presented in Appendix B.2. The HR-Agent makes a decision every $N$ steps, and the selected Hedger executes hedging at each step. **Transition**: The OP State contains market features $F_t$ only, so its transition $s_{t+1}^{\text{op}} = T^{\text{op}}(s_t^{\text{op}})$ is the market dynamics, which is not affected by the action. However, the positions and Greeks in $s_{t+1}^{\text{hr}}$ are affected by the two agents: the OP-Agent determines which options to hold, and the HR-Agent's selection of Hedger affects how the underlying position evolves under varying market conditions. Thus, the transition $s_{t+N}^{\text{hr}} = T_n^{\text{hr}}(s_t^{\text{hr}}, a_t^{\text{op}}, a_t^{\text{hr}})$ takes the action of two agents into account. **Reward function**: The reward for OP-Agent is defined as the change in portfolio net value $r_{t+1}^{\text{op}} = V_{t+1} - V_t$; the reward of HR-Agent is defined as the advantage of the selected Hedger $\pi_{a_t^{\text{hr}}}^{\text{hedger}}$ over a rule-based baseline Hedger $r_{t+n^{\text{hr}}}^{\text{hr}} = V_{t+n^{\text{hr}}} - \hat{V}_{t+n^{\text{hr}}}$.

## 4 OPHR: A Two-Phase Multi-Agent Framework for Volatility Trading

In this section, we demonstrate our OPHR framework as shown in Figure 2 to solve the volatility trading problem. We first introduce the training of the option part, consisting of an OP-Agent, which involves an n-step TD error update. Then we introduce the hedging part, which consists of a set of Hedgers with different risk preferences and an HR-Agent selecting the optimal Hedger based on positions and market conditions. To make the two parts work coordinately with each other, we design a joint training method, which first initializes the OP-Agent with a sub-optimal Oracle policy, then trains the HR-Agent and the OP-Agent alternatively.

## 4.1 OP-Agent Optimization

OP-Agent is responsible for determining the holdings of options at each time step. Its primary objective is to adjust the overall payoff structure of the portfolio as a function of the price of the underlying asset, according to current market conditions, enabling the agent to systematically capture profits from price fluctuations of the underlying asset.

**n-step TD Learning.** In options trading, portfolio returns are typically realized over time, as long Gamma strategies rely on large price movements, and short Gamma strategies accumulate theta over time. In addition, frequent hedging incurs transaction costs, introducing noise into short-term rewards. These issues render the reward signal both delayed and noisy, limiting the effectiveness of one-step TD updates to capture the true value of positioning decisions. To address this, we adopt n-step TD learning, which incorporates a sequence of future rewards and a bootstrapped value estimation, enabling the agent to better estimate the long-term value of current positioning decisions.

$$\mathcal{L}(\theta) = \mathbb{E}\left[\left(\sum_{k=0}^{n-1}\gamma^k r_j^{(k)} + \gamma^n Q_{\theta'}(s_j^{(n)}, \arg\max_a Q_\theta(s_j^{(n)}, a)) - Q_\theta(s_j, a_j)\right)^2\right] \tag{5}$$

The algorithm for online learning is presented in Algorithm 1.

---

**Algorithm 1: OP-Agent Online Training via $n^{\text{op}}$-step TD Error**

---

1: Reinitialize trading environment $Env$
2: **for** $t \in \text{Range}(0, T, n^{\text{op}})$ **do**
3:      **for** $t \in \text{Range}(0, n^{\text{op}}, 1)$ **do**
4:          **if** $t \mod n^{\text{hr}} == 0$ **then** Get HR Action
5:              $a_t^{\text{hr}} = \arg\max_a Q_\theta(s_t^{\text{hr}}, a)$, set $\pi_{a_t^{\text{hr}}}^{\text{hr}}$ as Hedger
6:          With probability $\epsilon$, choose a random action, otherwise $a_t^{\text{op}} = \arg\max_a Q_\theta(s_t^{\text{op}}, a^{\text{op}})$
7:          Execute actions and get next State $s_{t+1}^{op}$ and Reward $r_{t+1}^{op}$
8:      Store $n^{\text{op}}$-step transition: $(s_t, a_t^{\text{op}}, \{r_{t+1}^{\text{op}}, \dots, r_{t+n^{\text{op}}}^{\text{op}}\}, s_{t+n^{\text{op}}})$ in $\mathcal{R}^{\text{op}}$
9:      Sample a batch of transitions from $\mathcal{R}^{\text{op}}$ and update $\phi$ with $n^{\text{op}}$-step TD error Eq. (5)
10: **return** $Q_\phi$

---

## 4.2 HR-Agent Optimization with DT-Hedger Baseline

**Baseline Hedger.** As we mentioned in Section 2.2, DDH plays a vital role in volatility trading, which manages risk and realizes profit by adjusting the underlying position. The Delta threshold hedging strategy is a commonly used rule-based method that monitors the Delta exposure of the current portfolio and trades the hedging instruments to make the Delta exposure 0 when the Delta exposure exceeds the threshold. In this work, the Baseline Hedger $\hat{\pi}^{\text{hedger}}$ is used to compute the reward for the HR-Agent and in other baseline experiments.

**Deep Hedgers.** A key drawback of the Baseline Hedger is that it only considers the snapshot of $\Delta$ exposure. However, all Greeks ($\Delta, \Gamma, \Theta$ and Vega) are functions of time $t$ and IV $\sigma$ and the underlying price $S_t$, which change dynamically as time passes and the market evolves. Therefore, DRL-based methods have been applied to train better so-called Deep Hedgers to optimize the long-term performance. Murray et al. [2022] provides an actor-critic alternative to train a set of Deep Hedgers $\{\pi_{(i)}^{\text{hedger}}\}_{i=1}^{K}$ with different risk aversions on simulated data.

**Hedger-Routing Agent.** However, as illustrated in Figure 1, it is important to consider the market condition when making the hedging decisions, but the Hedgers do not take the market conditions into account, and it is difficult to incorporate the real-world market data into the training protocol of Deep Hedgers by simulation. To further optimize the hedging behavior, we employ the HR-Agent to select Deep Hedgers. The HR-Agent chooses an optimal Deep Hedger $\pi_{(a_t^{\text{hr}})}^{\text{hedger}}$ based on the portfolio and market conditions at every $n^{\text{hr}}$ step. We execute the selected Deep Hedger $\pi_{(a_t^{\text{hr}})}^{\text{hedger}}$ in $Env$ and the Baseline Hedger $\hat{\pi}^{\text{hedger}}$ in a twin environment, $\widehat{Env}$ duplicated from $Env$, for $n^{\text{hr}}$ steps. Then the reward for $a_t^{\text{hr}}$ is computed as the difference of net values in the two environments:

$r_{t+n^{hr}}^{hr} = V_{t+n^{hr}} - \hat{V}_{t+n^{hr}}$. This relative reward design helps the training of the HR-Agent. The training algorithm of the HR-Agent is demonstrated in Algorithm 2.

---

**Algorithm 2: HR-Agent Online Training via 1-step TD Error**

---

1: Initialize $Env$ and $\widehat{Env}$
2: **for** $t \in \text{Range}(0, T, n^{hr})$ **do**
3:     With probability $\epsilon$, choose a random action, otherwise $a_t^{hr} = \arg\max_a Q_\psi(s_t^{hr}, a^{hr})$
4:     **for** $t \in \text{Range}(0, n^{hr}, 1)$ **do**
5:         Get OP Action $a_t^{op} = \pi^{op}(s_t^{op})$
6:         Use selected Hedger $\pi_{(a_t^{hr})}^{hedger}$ in $Env$, and baseline Hedger $\hat{\pi}^{hedger}$ in $\widehat{Env}$
7:         Execute actions and get the next State $s_{t+1}^{op}$
8:     Get $r_{t+n^{hr}}^{hr}$, store $(s_t^{hr}, a_t^{hr}, r_{t+n^{hr}}^{hr}, s_{t+n^{hr}}^{hr})$ in $\mathcal{R}^{op}$, and synchronize $\widehat{Env}$ with $Env$
9:     Sample batch from $\mathcal{R}^{op}$ and perform DQN update $\psi$ with Eq. (5), soft update $\psi'$
10: **return** $Q_\psi$

---

### 4.3 OPHR Training

**Phase 1: Offline Initialization.** Long trajectories in n-step TD increase the variance of target returns, often leading to unstable training and slow convergence. So we employ a sub-optimal Oracle policy to generate initial experience for guidance in the initialization phase. The Oracle OP policy $\pi_{Oracle}^{op}$ generates long/short signals by comparing the future RV and current IV and employs the Baseline Hedger $\hat{\pi}^{hedger}$ to make hedging decisions. Specifically, if the future RV $\geq \sigma(1 + \beta)$ places a long position, if the future RV $\leq \sigma(1 - \beta)$ places a short position; otherwise, place a neutral position. While inherently sub-optimal, its action aligns with desirable and profitable trading behaviors, significantly reducing the exploration burden, accelerating convergence.

**Phase 2: Iterative Online Training.** To further improve the trading performance, we alternatively train the HR-Agent and OP-Agent. The necessities of iterative training lie in 2 folds. **i) From the MARL perspective:** iterative training can reduce complexity, improve stability and convergence. **ii) From the option trading perspective:** the OP-Agent can take more aggressive positions if the hedging policy is more sophisticated, and the HR-Agent also needs to learn how to handle these more aggressive positions. The detailed implementation is presented in Algorithm 3 in Appendix B.3.

## 5 Experiment

### 5.1 Experiments Setup

**Datasets.** To comprehensively evaluate the proposed algorithm, we conduct experiments on BTC and ETH options data obtained from Deribit. The dataset splitting is shown in Table 1. This period covers diverse market conditions, including bull markets (e.g., 2019 and 2021), bear markets (e.g., 2022), and periods of elevated volatility (e.g., the 2020 COVID-19 pandemic and the 2022 crypto market crash). We utilize hourly-level data to capture intraday price dynamics and volatility structures critical to volatility trading. The dataset includes complete options chains across a wide range of strikes and expirations (weekly to quarterly), as well as comprehensive market indicators such as implied volatility surfaces, open interest, and trading volume.

Table 1: Dataset Splits for Experiments

| Dataset | Train | Validation | Test |
|---------|-------|------------|------|
| BTCUSD | 19/04/01 – 22/12/31 | 23/01/01 – 23/06/30 | 23/07/01 – 24/07/01 |
| ETHUSD | 19/04/01 – 22/12/31 | 23/01/01 – 23/06/30 | 23/07/01 – 24/07/01 |

**Evaluation Metrics.** We evaluate our method using 8 financial metrics: 1 profit, 3 risk-adjusted profit, 2 risk and 3 trade criteria. Returns are aggregated into day-level before calculation. **Total Return (TR)** is the overall return rate of the test period. **Annual Volatility (AVOL)** is the standard deviation of daily returns annualized. **Maximum Drawdown (MDD)** measures the largest loss

from any peak to show the downside risk of the strategy. **Annual Sharpe Ratio (ASR)** is the profit adjusted by volatility risk. **Annual Calmar Ratio (ACR)** measures profit adjusted by downside risk. **Annual Sortino Ratio (ASoR)** applies downside deviation as the risk measure. **Win Rate (WR)** is the percentage of trades that result in a profit. **Profit/Loss Ratio (PLR)** measures the average profit of winning trades relative to the average loss of losing trades. **Holding Period (HP)** is the average holding time per trade. The detailed definitions of these metrics are presented in Appendix C.1.

Table 2: Performance comparison on 2 Crypto markets with 8 baselines. Pink, green, and blue results show the best, second-best, and third-best results.

| Data | Model | Profit | Risk-Adjusted Profit | | | Risk Metrics | | Trade Metrics | |
|------|-------|--------|------|------|-------|----------|----------|--------|------|
| | | TR(%)↑ | ASR↑ | ACR↑ | ASoR↑ | AVOL(%)↓ | MDD(%)↓ | WR(%)↑ | PLR↑ |
| BTC | Long | -33.05 | -1.32 | -0.77 | -2.13 | 24.90 | 42.70 | 21.74 | 1.82 |
| | Short | 2.90 | 0.09 | 0.10 | 0.09 | 24.55 | 21.10 | 73.91 | 0.37 |
| | MR | -19.80 | -1.32 | -0.74 | -1.54 | 15.40 | 27.45 | 41.94 | 1.18 |
| | MOM | -36.30 | -1.85 | -0.85 | -1.61 | 20.45 | 44.80 | 46.99 | 0.78 |
| | GARCH | -40.83 | -4.29 | -0.95 | -5.54 | 9.51 | 42.76 | 26.32 | 0.78 |
| | DeepVol | -14.24 | -1.30 | -0.65 | -2.44 | 10.92 | 21.85 | 36.36 | 1.33 |
| | GBDT | -30.45 | -2.88 | -0.91 | -4.75 | 11.22 | 35.25 | 35.16 | 1.15 |
| | MLP | -74.55 | -4.73 | -1.13 | -5.14 | 18.36 | 76.80 | 26.39 | 1.20 |
| | LSTM | -21.78 | -1.99 | -0.86 | -3.26 | 11.46 | 26.64 | 40.00 | 1.13 |
| | DLOT | 4.91 | 0.52 | 0.55 | 0.66 | 21.40 | 8.92 | 47.97 | 1.11 |
| | OP | 21.43 | 1.19 | 1.46 | 2.03 | 17.01 | 13.84 | 44.50 | 1.86 |
| | OPHR | 33.10 | 1.87 | 3.35 | 3.27 | 16.83 | 9.41 | 45.93 | 2.05 |
| ETH | Long | -28.25 | -0.72 | -0.53 | -0.83 | 37.55 | 50.95 | 34.78 | 0.71 |
| | Short | -34.75 | -1.12 | -0.69 | -1.25 | 33.00 | 53.05 | 56.52 | 0.48 |
| | MR | -19.80 | -1.32 | -0.74 | -1.54 | 15.40 | 27.45 | 41.94 | 1.18 |
| | MOM | -36.30 | -1.53 | -0.84 | -1.46 | 23.90 | 43.65 | 44.44 | 0.87 |
| | GARCH | -52.59 | -3.79 | -0.90 | -4.18 | 13.85 | 58.10 | 12.50 | 0.62 |
| | DeepVol | -22.82 | -1.79 | -0.96 | -3.03 | 12.71 | 23.76 | 42.86 | 1.15 |
| | GBDT | -48.54 | -3.76 | -1.06 | -4.55 | 14.22 | 50.28 | 30.00 | 0.98 |
| | MLP | -70.26 | -4.83 | -1.15 | -5.37 | 16.80 | 70.59 | 29.01 | 1.03 |
| | LSTM | -62.67 | -4.62 | -1.11 | -5.03 | 15.42 | 64.44 | 31.31 | 0.75 |
| | DLOT | 1.19 | 0.07 | 0.09 | 0.09 | 17.14 | 13.53 | 42.80 | 1.34 |
| | OP | 23.49 | 0.85 | 0.77 | 0.79 | 26.85 | 29.45 | 52.74 | 1.29 |
| | OPHR | 44.89 | 1.76 | 1.58 | 1.67 | 24.42 | 27.25 | 55.61 | 1.43 |

**Baselines.** To comprehensively evaluate our method, we implement multiple baseline approaches categorized into three major groups: **Directional Volatility Strategies:** These include pure **Long** Volatility and pure **Short** Volatility strategies. The Long Volatility strategy purchases at-the-money straddles when volatility conditions are favorable, while the Short Volatility strategy sells at-the-money straddles when IV exceeds historical RV. **Single Factor Models:** These strategies allocate long or short positions based on mean reversion (**MR**)[Poterba and Summers, 1988, Wong and Lo, 2009, Wood et al., 2022]or momentum (**MOM**)[Moskowitz et al., 2012, Lim et al., 2019, Wood et al., 2022, Tan et al., 2023, Heston et al., 2023, Wood et al., 2023] factors. **Machine Learning Models:** We implement **GBDT**[Dorogush et al., 2018], **MLP**[LeCun et al., 2015], and **LSTM**[Hochreiter and Schmidhuber, 1997], to predict optimal long/short volatility positions by leveraging their respective strengths in capturing non-linear relationships, complex feature interactions, and temporal dynamics in volatility patterns. We also include **GARCH**[Bollerslev, 1986], the seminal econometric model for volatility forecasting that captures time-varying conditional heteroskedasticity through autoregressive dynamics, serving as the classical benchmark in volatility prediction, and **DeepVol**[Moreno-Pino and Zohren, 2024], a deep learning framework for volatility forecasting, representing the current frontier in neural network-based volatility modeling. Additionally, we create an end-to-end DL baseline following **DLOT**[Tan et al., 2024], which applies deep learning to directly predict the options' PnL instead of volatility. **Our Method:** To evaluate the effectiveness of different parts of our method, **OP** uses the OP-Agent and the rule-based Hedger.

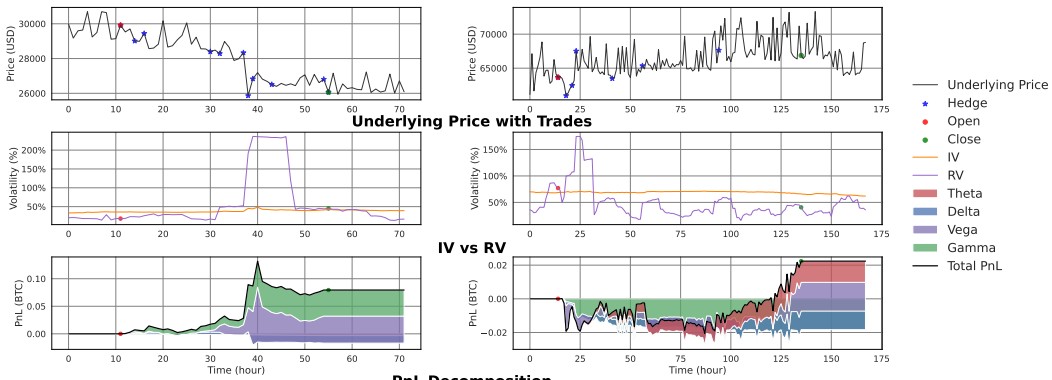

Figure 3: **Trade examples of OPHR on BTC.** The left panel illustrates a long $\Gamma$ trade, which primarily generates profits from $\Gamma$ and Vega exposure, while incurring theta costs. The right panel presents a short $\Gamma$ trade, where the strategy mainly profits from theta accrual over the holding period and gains from a decline in implied volatility, with the primary risk stemming from $\Gamma$ exposure.

## 5.2 Baseline Comparison

As demonstrated in Table 2, our proposed OPHR approach exhibits consistent superiority across both BTC and ETH markets. The method achieves the highest total returns while ranking first in all risk-adjusted performance metrics (ASR, ACR, and ASoR). These results underscore OPHR's effectiveness in risk management and profit optimization through its adaptive hedging framework.

In contrast, conventional directional volatility strategies display notable limitations due to their inherent tail risk profiles. Short Gamma positions achieve high WR but poor PLR because they collect small premiums consistently while remaining exposed to rare but catastrophic losses during volatility spikes. Conversely, Long Gamma strategies offer better PLR by capitalizing on large market movements, but suffer negative returns due to low WR and theta decay during market stability.

Traditional factor-based approaches (MR and MOM) demonstrate suboptimal performance, as they fail to adapt to abrupt volatility regime shifts that characterize cryptocurrency options markets. Similarly, ML baselines (GBDT, MLP, LSTM, DeepVol, GARCH) struggle with profitability despite occasionally achieving lower volatility, due to two fundamental limitations: first, they fail to bridge the gap between forecasting RV and optimizing path-dependent PnL outcomes in options trading; second, they consistently underperform in capturing the long-Gamma opportunities hidden in the fat tails of volatility distributions, precisely where the most significant profit potential exists, as discussed in Appendix D.2. The PnL curves are also presented in Figure 4 and 5 in the Appendix. DLOT significantly outperforms RV prediction-based methods, benefiting from an end-to-end design that avoids model assumptions - similar in spirit to our RL approach. However, lacking adaptive hedger selection and further RL optimization, it cannot match our method's performance.

## 5.3 Closer Look

Table 3: The long/short trading behaviour

| Data | Model | Long | | | Short | | |
|------|-------|-----|--------|------|-------|--------|------|
| | | HP | WR(%)↑ | PLR↑ | HP | WR(%)↑ | PLR↑ |
| BTC | OP | 20.94 | 31.96 | 2.21 | 52.94 | 55.86 | 1.60 |
| | OPHR | 21.29 | 33.68 | 2.28 | 51.12 | 56.64 | 1.87 |
| ETH | OP | 8.91 | 44.25 | 1.51 | 47.56 | 60.98 | 1.08 |
| | OPHR | 9.16 | 47.17 | 1.71 | 50.59 | 63.79 | 1.18 |

Table 3 reveals how the HR-Agent systematically enhances trading performance across position types. The distinct holding periods between long and short positions (approximately 9-21 hours for long versus 50-51 hours for short) reflect OPHR's sophisticated volatility timing capability. This

pattern aligns with the market reality that cryptocurrencies typically experience extended periods of relative stability (suitable for short volatility positions with longer holding periods), punctuated by brief episodes of extreme volatility (ideal for long volatility positions with shorter holding periods). OPHR effectively adapts to these regime shifts, maintaining short positions during calm periods while quickly capitalizing on and exiting from volatility spikes through tactical long positions.

This volatility timing ability translates directly into performance improvements, with OPHR enhancing WR by 1.72-2.92% for long positions and 0.78-2.81% for short positions compared to the base OP model. The PLR improvements are equally notable, particularly for short positions where OPHR's hedging refinements prevent catastrophic losses during unexpected volatility surges while minimizing unnecessary hedging costs during stable periods. These trade-level enhancements directly drive OPHR's superior overall performance metrics in Table 2, including higher total returns and significantly improved risk-adjusted metrics.

To further illustrate OPHR's trading strategies, we visualize representative trade examples in Figure 3. In a long $\Gamma$ position, OPHR actively hedges to capture $\Gamma$ and Vega gains, which typically outweigh theta costs, leading to positive returns. In contrast, the short $\Gamma$ position relies on theta accrual and benefits from declining implied volatility, while $\Gamma$ risk is managed through reduced hedging frequency. In both cases, the strategy balances risk and reward to achieve a net profit.

Table 4: Transaction costs as a percentage of total PnL of OPHR

| Strategy | Options Cost (%) | Underlying Cost (%) | Total Cost (%) |
|---|---|---|---|
| BTC | 4.15% | 5.21% | 9.36% |
| ETH | 2.97% | 2.78% | 5.75% |

**Transaction Cost Analysis.** OPHR's profitability remains robust after accounting for transaction costs. Table 4 shows that total transaction costs represent 9.36% and 5.75% of P&L for BTC and ETH respectively. We incorporate realistic commission fees following Deribit's fee structure: 0.05% for perpetual futures and 0.03% for options (capped at 12.5% of option price), with bid-ask spread costs implicitly captured through market order execution prices in backtesting.

These modest cost ratios result from OPHR's measured trading frequency, with average holding periods of 9-51 hours (Table 3) that minimize unnecessary transactions. The HR-Agent's sophisticated hedging decisions further reduce excessive rebalancing during stable periods while maintaining effective risk management during volatility spikes. This cost efficiency confirms that OPHR's superior performance reflects genuine alpha generation rather than unrealistic cost assumptions.

## 6   Conclusion

In this paper, we introduced OPHR, a novel reinforcement learning framework for volatility trading through options. Our OP-Agent excels at volatility timing by dynamically identifying market regimes where implied volatility is mispriced relative to expected RV. Complementing this, our HR-Agent optimizes performance by selecting appropriate hedging strategies that lock in profits during volatility spikes while minimizing costs during calmer markets. Empirical results on cryptocurrency options markets demonstrate that OPHR consistently outperforms traditional approaches across key metrics, validating that our multi-agent reinforcement learning framework effectively captures the complex patterns essential for successful volatility trading. Future work could explore additional volatility trading opportunities, such as volatility smile skewness and term structure anomalies, as well as further optimizing hedgers to incorporate market features for more precise hedging decisions. These enhancements would enable even more sophisticated volatility trading strategies while maintaining the data-driven, adaptive nature of our reinforcement learning framework.

## Acknowledgements

This research is supported by the National Research Foundation Singapore and DSO National Laboratories under the AI Singapore Programme (AISGAward No: AISG2-GC-2023-009).

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

# A Cryptocurrency Options

## A.1 The Black-Scholes-Merton Model and Greeks

The **Black-Scholes-Merton (BSM) Model** is a seminal framework for pricing European-style options. Despite the fact that real-world markets often violate its assumptions, the model offers fundamental insights and practical tools for derivatives pricing and risk management.

### A.1.1 Model Assumptions

The BSM model relies on several idealized assumptions:

- Markets are frictionless: there are no transaction costs or taxes.
- Trading is continuous, and investors can borrow and lend at a constant, risk-free interest rate $r$.
- The price of the underlying asset $S_t$ follows geometric Brownian motion, governed by the stochastic differential equation:

$$dS_t = \mu S_t dt + \sigma S_t dW_t,$$

where:

  - $\mu$ is the expected return of the asset,
  - $\sigma$ is the volatility of the asset,
  - $W_t$ is a standard Wiener process.

- There are no arbitrage opportunities.
- The option is European.

### A.1.2 Black-Scholes Pricing Formula

Under these assumptions, the value $V(t, S_t)$ of a European option satisfies the **Black-Scholes Partial Differential Equation (PDE)**:

$$\frac{\partial V}{\partial t} + \frac{1}{2}\sigma^2 S_t^2 \frac{\partial^2 V}{\partial S_t^2} + rS_t \frac{\partial V}{\partial S_t} - rV = 0.$$

Solving this PDE with appropriate boundary conditions yields closed-form pricing formulas for European call and put options:

$$C(t, S_t) = S_t N(d_1) - Ke^{-r(T-t)} N(d_2), \tag{6}$$

$$P(t, S_t) = Ke^{-r(T-t)} N(-d_2) - S_t N(-d_1), \tag{7}$$

where

$$d_1 = \frac{\ln(S_t/K) + \left(r + \frac{1}{2}\sigma^2\right)(T-t)}{\sigma\sqrt{T-t}}, \tag{8}$$

$$d_2 = d_1 - \sigma\sqrt{T-t}, \tag{9}$$

and $N(\cdot)$ is the cumulative distribution function of the standard normal distribution.

### A.1.3 Greeks

The BSM model also provides a set of sensitivity measures known as the **Greeks**, which quantify how the option price responds to changes in key variables:

- **Delta**: Measures sensitivity to the underlying price.

$$\Delta = \frac{\partial V}{\partial S}.$$

$$\Delta_{\text{call}} = N(d_1), \qquad \Delta_{\text{put}} = N(d_1) - 1.$$

  It is essential for directional risk management.

- **Gamma**: Measures the rate of change in Delta with respect to the underlying asset.

$$\Gamma = \frac{\partial^2 V}{\partial S^2}.$$

$$\Gamma = \frac{N'(d_1)}{S_t \sigma \sqrt{T-t}}$$

where $N'(d_1) = \frac{1}{\sqrt{2\pi}} e^{-d_1^2/2}$ is the standard normal probability density. High Gamma indicates that Delta changes rapidly with small movements in $S_t$.

- **Theta**: Measures the effect of time decay on option value.

$$\Theta = \frac{\partial V}{\partial t}.$$

$$\Theta_{\text{call}} = -\frac{S_t N'(d_1)\sigma}{2\sqrt{T-t}} - rKe^{-r(T-t)}N(d_2),$$

$$\Theta_{\text{put}} = -\frac{S_t N'(d_1)\sigma}{2\sqrt{T-t}} + rKe^{-r(T-t)}N(-d_2).$$

It is usually negative for long options, reflecting value loss over time.

- **Vega**: Measures sensitivity to volatility.

$$\text{Vega} = \frac{\partial V}{\partial \sigma}.$$

$$\text{Vega} = S_t N'(d_1)\sqrt{T-t}.$$

- **Rho**: Measures sensitivity to interest rate changes.

$$\rho_{\text{call}} = K(T-t)e^{-r(T-t)}N(d_2),$$

$$\rho_{\text{put}} = -K(T-t)e^{-r(T-t)}N(-d_2).$$

## A.2 Inverse Option

Inverse options are European-style, cash-settled option contracts denominated in the underlying cryptocurrency (e.g., BTC or ETH), representing the dominant format in contemporary crypto derivatives markets. These contracts can only be exercised automatically at expiry, eliminating the possibility of early exercise. Instead of requiring the physical delivery of the underlying asset, settlement occurs in the form of a payout denominated in the base cryptocurrency, calculated as the difference between the delivery price and the strike price. Option pricing is based on the Black-Scholes framework, wherein the forward price, rather than the spot index, is employed as the principal input for valuation. Trading is generally conducted on a continuous 24/7 basis and accommodates a diverse range of order types. Furthermore, the system incorporates risk management measures, including price correction mechanisms, to enhance market integrity and operational resilience. Inverse options are particularly well-suited for market participants seeking to hedge or speculate on crypto-asset exposures without engaging in spot or fiat transactions.

## A.3 Portfolio Margin

Traditional margin systems typically apply fixed margin requirements to individual positions, without accounting for the potential offsetting risk of multi-leg strategies. For instance, in short straddle positions—where both a call and a put option are sold with the same strike price and expiration—the worst-case loss is expected to occur in only one direction of price movement, not both simultaneously. Charging full margin on each leg independently leads to a substantial overestimation of portfolio risk, thereby reducing capital efficiency.

To address this inefficiency, the Portfolio Margin (PM) framework adopts a risk-based approach that estimates the required margin based on the portfolio's net risk exposure. Instead of evaluating each position in isolation, PM performs a series of stress tests on the entire portfolio, simulating potential profit and loss (P&L) under a range of underlying asset price and implied volatility scenarios. The maximum projected loss from these scenarios is then used as the basis for the maintenance margin requirement.

**Calculation Procedure.** The PM system simulates the portfolio P&L under multiple discrete asset price shifts (e.g., $-15\%, -12\%, \ldots, 0\%, \ldots, +15\%$) and volatility adjustments (e.g., $+45\%$, $0\%$, $-30\%$). For each price scenario $X\%$, the total portfolio P&L is computed as:

$$L(X\%) = A(X\%) + B(X\%)$$

where $A(X\%)$ represents the P&L of futures or perpetual contracts, and $B(X\%)$ is the worst-case loss of the option positions under the three volatility assumptions at the given price level. The **maximum potential loss (ML)** is determined as:

$$ML = \min\{L(X\%) \text{ for all scenarios}\}$$

An additional contingency margin is added to account for uncovered short options and large directional exposure:

$$\text{Contingency} = \left[\left(\sum \text{Net Short Options at Strike} \times a\%\right) + (|\text{Futures Position}| \times b\%)\right] \times \text{Spot Price}$$

Finally, the total maintenance margin is given by:

$$\text{Maintenance Margin} = |ML| + \text{Contingency}$$

This methodology ensures that the margin reflects the true tail risk of the portfolio while improving capital efficiency for hedged and risk-reducing strategies.

**Margin Control in Simulation Environment** In our experiments, the simulated trading environment adopts a Portfolio Margin (PM) framework to estimate margin requirements. For all conducted trading episodes, the required margin was constrained to remain below a fixed threshold of $50\%$ of the agent's total cash holdings. This constraint ensures the feasibility and solvency of all trading actions throughout the simulation, preventing excessive leverage and guaranteeing compliance with realistic capital limitations.

# B  Method

## B.1  Oracle OP-policy Design

As introduced in Section 4.1, the Oracle OP-policy, denoted as $\pi_{\text{Oracle}}^{\text{op}}$, is designed to provide structured supervision for reinforcement learning agents by leveraging privileged information—specifically, the future RV. This section presents the detailed implementation of the policy.

The strategy generates trading signals by comparing the future RV with the current IV, $\sigma_{\text{imp}}$, subject to a predefined tolerance threshold $\beta$. Formally, the policy operates as follows:

- If $\text{RV}_{\text{future}} \geq \sigma_{\text{imp}}(1 + \beta)$, then initiate a **long** position.
- If $\text{RV}_{\text{future}} \leq \sigma_{\text{imp}}(1 - \beta)$, then initiate a **short** position.
- Otherwise, maintain a **neutral** position.

In practice, the Oracle OP-policy serves as an oracle demonstrator by generating high-quality sequential action-value trajectories through direct interaction with the environment. When filling the experience buffer, the Oracle OP-policy utilizes future RV information $\text{RV}_{\text{future},i}$ across multiple time horizons, along with varying tolerance thresholds $\beta_i$, to construct a diverse set of oracle strategies. This diversity facilitates better coverage of the policy space and enables the OP agent to approximate near-optimal behavior more effectively during the early stages of training.

While inherently sub-optimal due to the unrealistic assumption of future information availability, this rule-based strategy aligns with desirable trading behaviors and effectively serves as a structured demonstration policy. By doing so, it reduces the exploration space and helps the learning agent to converge more rapidly toward profitable regions of the policy space. Hedging decisions under this policy are executed using a predefined baseline Hedger $\hat{\pi}_{\text{Hedger}}$.

It is important to note that, although the Oracle OP-policy utilizes future RV from the training set—a type of privileged information unavailable during deployment—this signal is used solely to generate offline trajectories for training purposes. These trajectories serve as high-quality demonstrations to

guide the learning process. Thus, no future information is exposed to the agent during inference, ensuring the integrity of forward-looking decision-making. As such, the policy improves sample efficiency and accelerates convergence without compromising generalization to out-of-sample, real-world environments.

Fundamentally, the Oracle OP policy functions as a proxy for an optimal strategy based on historical data, providing structured supervision in the form of offline expert demonstrations. Similar to methods in imitation and offline reinforcement learning, its purpose is to aid training rather than serve as a deployable policy. Since all supervision derives from past data, the approach remains consistent with realistic deployment constraints.

## B.2  Hedgers

To hedge the $\Delta$ risk of the position and to lock in $\Gamma$ profits, we employ a series of hedging instruments. These Hedgers are constructed using various methods to adapt to different market conditions. We simulate the use of three types of hedging instruments under different market regimes and position directions. Based on the overall return and risk exposure, we select the optimal Hedgers to construct a Hedgers pool, which serves as the candidate set for the HR-agent's hedging decisions.

- **Delta-based Hedger**: This type of Hedger determines whether to hedge based on the $\Delta$ value of the current position. A decision threshold $\Delta_{\text{thres}}$ is predefined. When the absolute $\Delta$ of the position exceeds this threshold,

$$|\Delta_t| > \Delta_{\text{thres}},$$

  a full Delta hedge is executed to neutralize the position's $\Delta$ exposure.

- **Price-based Hedger**: This Hedger triggers a hedge based on significant price movements. Let $P_t$ denote the current underlying price and $P_{\text{last hedge}}$ the underlying price at the last hedge. A threshold $P_{\text{thres}}$ is used to determine whether a new hedge is necessary. When the relative price change satisfies

$$\left| \frac{S_t}{S_{\text{last hedge}}} - 1 \right| > S_{\text{thres}},$$

  a full hedge is performed to adjust the position accordingly.

- **Deep Hedgers**: This class of Hedgers leverages deep reinforcement learning to learn adaptive hedging strategies in a data-driven manner. Unlike rule-based Hedgers, deep Hedgers dynamically determine hedging actions by interacting with the environment, optimizing for a risk-adjusted objective. Specifically, we use the Actor-Critic algorithm to train these agents under various market regimes and position profiles.

  Each Deep Hedger receives the current market state as input, which includes the underlying price, option Greeks, historical price volatility, and the current position. The output is a continuous hedging action between 0 and 1, representing the proportion of $\Delta$ exposure to hedge.

  The training objective balances hedging cost and residual portfolio risk through a utility-based loss:

$$\mathcal{L}_{\text{hedge}} = -R_t + \frac{1}{\lambda} \log \mathbb{E}[e^{-\lambda X_t}],$$

  where $R_t$ is the immediate hedging reward (typically negative cost), $X_t$ is the wealth with post-hedging residual risk, and $\lambda$ is the risk-aversion parameter. A larger $\lambda$ encourages more conservative (risk-sensitive) behavior. By training under different simulated market conditions and position directions, Deep Hedgers learn to generalize across regimes and exhibit context-aware hedging behavior.

- **Hedger Pool Construction**: To enhance the HR-agent's hedging effectiveness across diverse market regimes, we construct a comprehensive Hedger pool composed of the three aforementioned types of Hedgers: Delta-based Hedgers, Price-based Hedgers, and Deep Hedgers.

  We generate a diverse set of Delta-based and Price-based Hedgers by varying their respective decision thresholds. Meanwhile, Deep Hedgers are trained under a range of simulated market

environments, position direction, and risk-aversion parameters $\lambda$. The resulting collection of Hedgers is then evaluated via backtesting across segmented market regimes—categorized by volatility levels and directional dynamics (e.g., trending vs. mean-reverting)—and position types (e.g., long $\Gamma$ or short $\Gamma$).

To ensure diversity in hedging behaviors and risk exposures, we retain only the top-$k$ performing Hedgers that achieve a favorable trade-off between hedging cost and risk control, primarily in terms of $\Delta$ exposure. These selected Hedgers collectively form the candidate Hedger pool, which serves as the basis choices for dynamic Hedger selection by the HR-agent within the full framework. This enables the construction of diverse and efficient hedging strategies to adapt to varying market conditions.

### B.3   Joing Training

---

Algorithm 3: OPHR Joint Training

---

**Require:** Oracle OP-Policy $\pi_{\text{Oracle}}^{\text{op}}$, Baseline Hedger $\hat{\pi}^{\text{Hedger}}$, $Env$ and Twin $\widehat{Env}$
 1: Initialize replay buffer $\mathcal{R}^{\text{op}}$, $\mathcal{R}^{\text{hr}}$, Q-networks $Q_\phi$, $Q_\psi$ and target Q-networks $Q_{\phi'}$, $Q_{\psi'}$
 2: **Phase 1: OP-Agent Offline Pretraining with Oracle OP-Policy and Baseline Hedger**
 3: Collect trajectories using $\pi_{\text{Oracle}}^{\text{op}}$ and $\hat{\pi}^{\text{Hedger}}$, store transitions in $\mathcal{R}^{\text{op}}$
 4: **for** Offline Training Iterations **do**
 5:     Sample a batch of transitions from $\mathcal{R}^{\text{op}}$ and update $\phi$ with $n^{\text{op}}$-step TD error Eq. (5)
 6: **Phase 2: Joint Online Learning**
 7: **for** Joint Training Epochs **do**
 8:     Fix $Q_\phi$ and train HR-Agent $Q_\psi$ using Algorithm 2 for $N^{\text{hr}}$ episodes
 9:     Fix $Q_\psi$ and train OP-Agent $Q_\phi$, using Algorithm 1 for $N^{\text{op}}$ episodes
10: **return** $Q_\phi$, $Q_\psi$

---

## C   Experiment

### C.1   Evaluation Metrics.

We evaluate our method using 8 financial metrics: 1 profit, 3 risk-adjusted profit, 2 risk and 2 trade criteria. Returns are aggregated into day-level before calculation.

- **Total Return (TR)** is the overall return rate of the test period, defined as $TR = \frac{V_t - V_1}{V_1}$, where $V_t$ is the final margin balance and $V_1$ is the initial margin balance.

- **Annual Volatility (AVOL)** is the the standard deviation of daily returns annualized defined as $\sigma[\mathbf{ret}] \times \sqrt{m}$ to measure the volatility risk, where $\mathbf{ret} = (ret_1, ret_2, ..., ret_t)$ is a vector of daily return, $\sigma[.]$ is the standard deviation function, and m is the annualization factor 365.

- **Maximum Drawdown (MDD)** measures the largest loss from any peak to show the downside risk of the strategy.

- **Annual Sharpe Ratio (ASR)** is the profit adjusted by volatility risk, defined as: $SR = \frac{E[\mathbf{ret}]}{\sigma[\mathbf{ret}]} \times \sqrt{m}$, where $E[\mathbf{ret}]$ is the expectation of daily return.

- **Annual Calmar Ratio (ACR)** is defined as $ACR = \frac{E[\mathbf{ret}]}{MDD} \times m$, measuring profit adjusted by downside risk.

- **Annual Sortino Ratio (ASoR)** applies downside deviation as the risk measure. It is defined as: $ASoR = \frac{E[\mathbf{ret}] \times \sqrt{m}}{DD}$, where downside deviation(DD) is the standard deviation of the negative daily return rates.

- **Win Rate (WR)** is the percentage of trades that result in a profit, defined as $WR = \frac{N_{win}}{N_{total}} \times 100\%$, where $N_{win}$ is the number of profitable trades and $N_{total}$ is the total trades.

- **Profit/Loss Ratio (PLR)** measures the average profit of winning trades relative to the average loss of losing trades, defined as $PLR = \frac{|E[\mathbf{ret}_{win}]|}{|E[\mathbf{ret}_{loss}]|}$, where $E[\mathbf{ret}_{win}]$ is the average return of profitable trades and $E[\mathbf{ret}_{loss}]$ is the average return of losing trades.

- **Holding Period (HP)** is the average holding time per trade, defined as $HP = \frac{\sum_{i=1}^{N_{total}} h_i}{N_{total}}$, where $h_i$ is the holding period of the $i$-th trade and $N_{total}$ is the total number of trades. It reflects the trading frequency.

## C.2 Hyperparameters

In Section 4.1, we introduce the N-step temporal-difference Double DQN algorithm applied within a rolling training framework. The main hyperparameter settings for this algorithm are summarized in Table 5. In our experiments, we adopt a rolling training of every 10 days and apply the proposed algorithm to historical data, and the network is updated using 12-step temporal-difference (TD) learning.

Table 5: Hyperparameters for OP-Agent Training

| Parameter | Description | Value |
|---|---|---|
| Training window size | Training window size | 10 |
| Oracle future RV | Time horizons for Oracle future RV | (3, 6, 9, 12, 24) |
| Oracle threshold $\beta$ | Thresholds for oracle policy | (0.1, 0.2, 0.4, 0.6, 0.8) |
| Oracle exploration rate $\epsilon_{\text{oracle}}$ | Oracle exploration rate | 0.1 |
| Episodes | Number of episodes per training window | 20,000 |
| Updates per episode | Network updates after each episode | 20 |
| Batch size | Number of samples per training batch | 512 |
| Hidden layer size | Maximum hidden layer size in OP-Agent | 1024 |
| Learning rate $\alpha$ | Step size for gradient updates | $1 \times 10^{-4}$ |
| N-step TD $n$ | Steps used in n-step TD learning | 12 |
| Discount factor $\gamma$ | Reward discount factor | 0.99 |
| Soft update coefficient $\tau$ | Rate for target network soft updates | 0.005 |
| Target update frequency | Frequency of target network updates | 10 |
| Dropout rate | Dropout rate in network layers | 0.2 |
| $\epsilon_{\text{start}}$ | Initial exploration rate | 0.9 |
| $\epsilon_{\text{end}}$ | Final exploration rate | 0.01 |
| $\epsilon_{\text{decay}}$ | Decay rate of $\epsilon$ (steps) | 10,000 |

As described in Section B.2, we utilize three types of Hedgers, each trained under different market regimes and position conditions. For each setting, multiple Hedgers are trained, and the top 30 performing ones are selected based on validation performance. These selected models serve as candidate Hedgers for the HR-agent. The hyperparameter settings for the Hedgers are summarized in Table 6, while the preliminary training parameters for the HR-agent are listed in Table 7.

Table 6: Hedger parameters.

| Hedger type | Parameter values |
|---|---|
| Delta-based Hedger | 0, 0.01, 0.02, 0.04, 0.06, 0.08, 0.1, 0.12, 0.15, 0.2, 0.25, 0.3, 0.35, 0.4, 0.5, 0.6, 0.7, 0.8, 0.9 |
| Price-based Hedger | 0.5%, 1%, 2%, 3%, 4%, 5%, 7%, 10%, 12%, 15%, 20%, 30%, 50%, 100% |
| Risk Aversion parameter $\lambda$ | 0.1, 0.2, 0.5, 0.7, 1, 2, 3, 5, 8, 10, 20, 40, 60, 80, 100, 200, 400, 800 |
| Hedger pool size | 30 |

Table 7: Hyperparameters for HR-Agent Training

| Parameter | Description | Value |
|---|---|---|
| Learning rate $\alpha$ | Step size for Q-network updates | $1 \times 10^{-4}$ |
| Discount factor $\gamma$ | Future reward discount factor | 0.99 |
| Batch size | Number of samples per update | 512 |
| Hidden layer size | Size of each hidden layer in the Q-network | 1024 |
| $\epsilon_{\text{start}}$ | Initial exploration probability | 0.9 |
| $\epsilon_{\text{end}}$ | Final exploration probability | 0.01 |
| $\epsilon_{\text{decay}}$ | Steps over which $\epsilon$ decays linearly | 10,000 |
| Updates per step | Number of Q-network updates per routing step | 10 |
| Routing interval $n_{\text{hr}}$ | Steps between Hedger selections | 24 |

**Training Setup.** We conducted all experiments on a server equipped with 4 NVIDIA RTX 4090 GPUs and an AMD Ryzen Threadripper PRO 5995WX CPU. The total time required for one iteration of stage 1 and stage 3 was approximately 12 hours for the OP-Agent optimization in Section 4.1, and around 3 hours for training the HR-agent in Section 4.2.

## C.3 Baselines

This section presents the detailed implementation of the baselines introduced in Section 5.1. All baseline strategies adhere to the following fundamental principles:

- **Open Position:** At the time of initiation, the strategy selects the nearest-to-the-money (ATM) straddle within a predefined maturity range as the underlying instrument. The notional size of the position is determined based on the allocated margin usage.

- **Close Position:** Depending on the nature of the signal, a strong reversal signal may indicate either a position reversal or position closure. Alternatively, for certain signals, a take-profit and stop-loss rule is applied: the position is closed if the profit reaches $p\%$ or if the loss reaches $l\%$. In addition, a maximum holding period $HP_{\max}$ is imposed; if neither take-profit nor stop-loss conditions are met before $HP_{\max}$, the position is force-closed upon reaching this time limit.

- **Hedger:** For consistency across all baselines, we apply a unified Delta-based hedging scheme as described in Section B.2, where the hedging threshold is set to $\Delta_{\text{thres}} = 0.1$.

### C.3.1 Directional Volatility Strategies

This strategy involves taking directional exposure to volatility by trading at-the-money (ATM) straddles. Specifically, straddles with time to maturity between $m_{\min}$ and $m_{\max}$ are selected, and each position is held until a fixed rollover point $m_{\text{rollover}}$. At each rollover date, existing positions are closed, and new positions are established following the same maturity selection criteria.

Two directional variants are considered:

- *Long*: Buys ATM straddles that meet the maturity condition and holds the position until the rollover date.

- *Short*: Sells ATM straddles under the same maturity condition and holds the position until the rollover date.

A unified rule-based Delta Hedger is applied to this strategy.

All parameters used in this strategy are summarized in Table 8.

### C.3.2 Single Factor Models

Both the Mean Reversion (MR) and Momentum (MOM) strategies are constructed based on a single indicator derived from the percentile of realized volatility (RV).

Table 8: Parameter settings for Long and Short Volatility strategies

| Parameter | Long Volatility | Short Volatility |
|---|---|---|
| $m_{\text{max}}$ | 90 | 90 |
| $m_{\text{min}}$ | 60 | 60 |
| $m_{\text{rollover}}$ | 21 | 21 |
| $\Delta_{\text{thres}}$ | 0.1 | 0.1 |

**RV Percentile**    For each evaluation date, the model computes the annualized realized volatility over a backward-looking window of length $P_{\text{vol}}$, denoted by $\sigma_{\text{p}}$. This RV $\sigma_{\text{p}}$ is then compared against the historical distribution of RV computed over a rolling lookback window of length $P_{\text{lookback}}$, yielding a percentile score.

Based on this percentile:

- *Mean Reversion (MR):* Takes a short volatility position when the current $\sigma_{\text{real}}$ percentile is high (expecting reversion), and a long volatility position when it is low.

- *Momentum (MOM):* Takes a long volatility position when the percentile momentum is high (expecting continuation), and a short volatility position when it is low. Specifically, the momentum of the percentile score is defined as the relative change over a lag of $i$ periods:

$$\text{Mo}_{\text{percentile}, i} = \left( \frac{\text{Percentile}_t}{\text{Percentile}_{t-i}} - 1 \right) \times 100\% \qquad (10)$$

Positions are opened by selecting at-the-money (ATM) straddles with maturities between $m_{\text{min}}$ and $m_{\text{max}}$, conditional on the factor signal.

Position closure follows the general rule-based principles described in Section 5.1 .

A unified rule-based Delta Hedger is applied to this strategy.

All parameters used in this strategy are summarized in Table 9.

Table 9: Parameter settings for Single Factor Models (Mean Reversion and Momentum)

| Parameter | MR | MOM |
|---|---|---|
| $P_{\text{vol}}$ (window for realized volatility) | 3, 6, 9, 12, 24 hours | |
| $P_{\text{lookback}}$ (lookback window for percentile) | 365 days | |
| $m_{\text{min}}, m_{\text{max}}$ (option maturity range) | 60, 90 | |
| $\Delta_{\text{thres}}$ (Delta hedging threshold) | 0.1 | |
| $p$ (take-profit threshold) | 5% | |
| $l$ (stop-loss threshold) | 3% | |
| $HP_{\text{max}}$ (maximum holding period) | 96 hours | |
| $S_{\text{MR,long}}, S_{\text{MR,short}}$ (MR threshold) | 20, 80 | – |
| $S_{\text{MOM,long}}, S_{\text{MOM,short}}$ (MOM threshold) | – | 15%, –10% |

### C.3.3  Machine Learning Models

We implement three supervised learning models to generate volatility forecasts: Gradient Boosted Decision Trees (GBDT), Multi-Layer Perceptron (MLP), and Long Short-Term Memory networks (LSTM).

Each model is trained to predict the annualized RV over a forward-looking window of length $P_{\text{vol}}$, which serves as the target variable. Input features include historical volatility measures, option market variables, and macroeconomic indicators characterizing the current market environment.

The predicted volatility is then compared to the current implied volatility (IV) of the ATM straddle to form a relative signal. The corresponding thresholds are selected based on backtesting, using values that yield the best performance on the training and validation sets.A rule-based decision rule is applied based on their ratio:

- A long straddle position is opened if the predicted volatility exceeds the current implied volatility by a ratio greater than $S_{\text{ML,long}}$, i.e., $\frac{\hat{\sigma}_{\text{pred}}}{\text{IV}_{\text{ATM}}} > S_{\text{ML,long}}$.
- A short straddle position is opened if the predicted volatility is sufficiently below the current implied volatility, i.e., $\frac{\hat{\sigma}_{\text{pred}}}{\text{IV}_{\text{ATM}}} < S_{\text{ML,short}}$.

Positions are opened using at-the-money (ATM) straddles with maturities between $m_{\text{min}}$ and $m_{\text{max}}$. Position closure follows the same rule-based framework described in Section 5.1:

- Take-profit is triggered when return exceeds $p = 5\%$;
- Stop-loss is triggered when loss exceeds $l = 3\%$;
- Maximum holding period: $HP_{\text{max}} = 96$ hours;
- If predicted volatility crosses the opposite threshold, the position is closed and reversed.

Delta risk is managed using the unified rule-based Delta hedging mechanism, with threshold $\Delta_{\text{thres}} = 0.1$.

All parameters for the machine learning strategies are summarized in Table 10.

Table 10: Parameter settings for Machine Learning Models

| Parameter | GBDT | MLP | LSTM |
|---|---|---|---|
| $P_{\text{vol}}$ | | (3, 6, 9, 12, 24)hours | |
| $m_{\text{min}}, m_{\text{max}}$ | | 60, 90 | |
| $\Delta_{\text{thres}}$ | | 0.1 | |
| $p$ | | 5% | |
| $l$ | | 3% | |
| $HP_{\text{max}}$ | | 96 hours | |
| $S_{\text{ML,long}}, S_{\text{ML,short}}$ | 1.6, 0.7 | 1.2, 0.9 | 1.3, 0.7 |
| Model architecture | CatBoost iterations=3000 depth=8 target='ic' | MLP max_hidden_dim=1024 depth=5 ReLU, dropout=0.2 | LSTM layers=3 max_hidden_dim=512 dropout=0.2 |

# D    More detailed Result Analysis

## D.1    Baseline Comparison

The visualization of the baseline comparison, summarized in Table 2, is presented in Figures 4 and 5.

## D.2    ML prediction

Table 11: Performance of ML Forecasting Models

| Model | BTC | | | ETH | | |
|---|---|---|---|---|---|---|
| | Spearman IC | Pearson IC | $R^2$ | Spearman IC | Pearson IC | $R^2$ |
| LSTM | 0.6133 | 0.6136 | 0.5776 | 0.5946 | 0.6192 | 0.5846 |
| MLP | 0.6030 | 0.4303 | 0.3353 | 0.5801 | 0.3306 | 0.5023 |
| CatBoost(GBDT) | 0.6214 | 0.6279 | 0.5781 | 0.6056 | 0.6152 | 0.5407 |

Table 11 presents the volatility forecasting performance of the baseline machine learning models. From the perspective of information coefficients (IC), all models generally demonstrate reasonably good predictive ability.

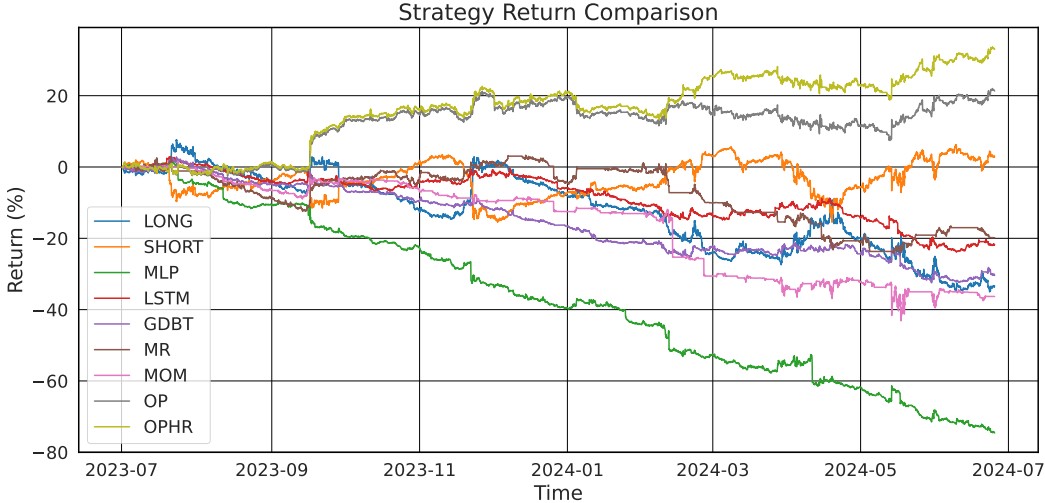

Figure 4: Baseline Comparison on BTC

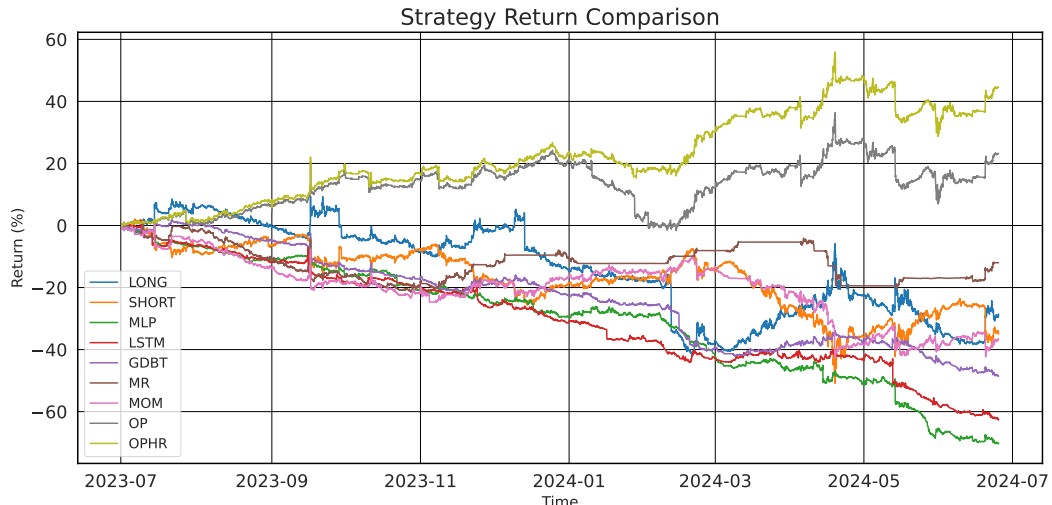

Figure 5: Baseline Comparison on ETH

Table 12: Performance of ML Forecasting Models on Top 5% Outliers

| Model | BTC | | | ETH | | |
|---|---|---|---|---|---|---|
| | Spearman IC | Pearson IC | $R^2$ | Spearman IC | Pearson IC | $R^2$ |
| LSTM | 0.3519 | 0.4460 | 0.1459 | 0.3462 | 0.5041 | 0.3363 |
| MLP | 0.3770 | 0.1903 | -0.8157 | 0.4101 | 0.4018 | -0.0536 |
| CatBoost(GBDT) | 0.3902 | 0.4746 | 0.2557 | 0.3664 | 0.5083 | 0.3578 |

Table 13: Performance of ML Forecasting Models on 95% Inliers

| Model | BTC | | | ETH | | |
|---|---|---|---|---|---|---|
| | Spearman IC | Pearson IC | $R^2$ | Spearman IC | Pearson IC | $R^2$ |
| LSTM | 0.6207 | 0.6619 | 0.6695 | 0.6032 | 0.6599 | 0.6483 |
| MLP | 0.6172 | 0.6633 | 0.6378 | 0.5894 | 0.6531 | 0.6050 |
| CatBoost(GBDT) | 0.6285 | 0.6700 | 0.6268 | 0.6158 | 0.6599 | 0.5783 |

However, these models fail to accurately capture the sharp spikes in realized volatility (RV), which are critical for long Gamma trading strategies. Table 12 shows that model performance drops noticeably on the top 5% outliers, reflecting limited robustness under extreme conditions. Moreover, there remains a substantial gap between predicting volatility and executing profitable trading decisions based on that prediction. Addressing this gap is precisely where our proposed method offers a distinct advantage.

### D.3 PNL Decomposition

To understand the sources of profit and loss (PnL) in option trading strategies, we decompose the cumulative PnL into analytically interpretable components based on the Greeks: Delta ($\Delta$), Gamma ($\Gamma$), Theta ($\Theta$), Vega, and a residual term capturing unexplained effects. Let $t = 1, \ldots, T$ denote discrete trading timestamps. The total PnL at time $t$ is approximated as:

$$\text{PnL}_t = \text{Delta}_t + \text{Theta}_t + \text{Vega}_t + \text{Gamma}_t + \text{Residual}_t \tag{11}$$

Each component corresponds to a specific source of risk exposure and is computed as follows:

**Delta PnL.** The contribution from changes in the underlying price $S_t$ is estimated by:

$$\text{Delta}_t = \Delta_{t-1} \cdot (S_t - S_{t-1}) \tag{12}$$

Here, $\Delta_{t-1} = \frac{\partial V}{\partial S}$ is the Delta exposure at the previous time step. This term captures the linear sensitivity of the option's value to movements in the underlying.

**Theta PnL.** The time decay of the option value is given by:

$$\text{Theta}_t = \Theta_{t-1} \cdot \Delta t \tag{13}$$

where $\Theta_{t-1} = \frac{\partial V}{\partial t}$ is the time sensitivity, and $\Delta t$ is the time step.

**Vega PnL.** The impact of changes in implied volatility $\sigma$ is estimated as:

$$\text{Vega}_t = \text{Vega}_{t-1} \cdot (\sigma_t - \sigma_{t-1}) \tag{14}$$

where $\text{Vega}_{t-1} = \frac{\partial V}{\partial \sigma}$ is the option's sensitivity to implied volatility.

**Gamma PnL (Realized).** Gamma measures the convexity of the option value with respect to the underlying. Between two hedge timestamps $t_{\text{prev}}$ and $t_{\text{curr}}$, the cumulative realized Gamma PnL is:

$$\text{Gamma}_{t_{\text{curr}}}^{\text{realized}} = \sum_{t=t_{\text{prev}}+1}^{t_{\text{curr}}} \frac{1}{2} \cdot \bar{\Gamma}_t \cdot (S_t - S_{t-1})^2 \tag{15}$$

where $\bar{\Gamma}_t = \frac{1}{2}(\Gamma_t + \Gamma_{t-1})$ is the average $\Gamma$ exposure at time $t$.

**Gamma PnL (Unrealized).** To monitor latent convexity risk, the unrealized Gamma PnL from the last hedge point to current time $t$ is tracked as:

$$\text{Gamma}_t^{\text{unrealized}} = \sum_{t'=t_{\text{last hedge}}+1}^{t} \frac{1}{2} \cdot \bar{\Gamma}_{t'} \cdot (S_{t'} - S_{t'-1})^2 \tag{16}$$

This quantity is not included in total PnL but helps visualize the risk of delayed hedging.

**Residual PnL.** The residual term accounts for the portion of PnL not explained by the above Greeks:

$$\text{Residual}_t = \text{PnL}_t - (\text{Delta}_t + \text{Theta}_t + \text{Vega}_t + \text{Gamma}_t) \tag{17}$$

Residual PnL may arise from several sources: model misspecification, transaction costs, bid-ask spreads, slippage during rebalancing, inaccurate Greek estimates, or discrete hedging errors. A high residual may indicate imperfect model assumptions or unmodeled market factors.

Table 14: PnL attribution (%) for OPHR on BTC and ETH

| Component | BTC Long | BTC Short | ETH Long | ETH Short |
|---|---|---|---|---|
| Delta | 1.17 | -0.55 | -5.26 | -23.50 |
| Gamma | 91.41 | -169.15 | 107.60 | -125.70 |
| Theta | -5.71 | 66.15 | -4.99 | 44.23 |
| Vega | 64.58 | 123.43 | 32.01 | 51.93 |
| Residual | -51.45 | 80.12 | -29.36 | 153.03 |

As shown in Table 14, we decompose the PnL of OPHR's long and short $\Gamma$ trading strategies into Greeks PNL and express each as a percentage of the total profit. Despite the presence of sizable residual terms in the attribution, the analysis reveals that the primary sources of profit in the long $\Gamma$ strategy stem from $\Gamma$ and Vega exposure, while the main cost is associated with Theta decay. In contrast, the short $\Gamma$ strategy benefits predominantly from Theta and Vega gains, while incurring substantial losses from adverse $\Gamma$ exposure. These observations align well with the intended design of the respective strategies.

# E   Broader Impact

Our reinforcement learning framework for options trading offers potential positive impacts through improved market liquidity, more accurate risk pricing, and democratization of sophisticated trading strategies. However, we acknowledge potential negative consequences including market concentration that could exacerbate inequality, possible herding behavior leading to market instability if widely adopted, and environmental concerns related to cryptocurrency markets where our method is tested. We emphasize the importance of responsible deployment to ensure algorithmic advances in financial markets benefit society broadly rather than concentrating advantages among a small group of sophisticated participants.

