# OpenReview forum: "OPHR: Mastering Volatility Trading with Multi-Agent Deep Reinforcement Learning"
_NeurIPS.cc/2025/Conference — NeurIPS 2025 poster_

### Official Review · Reviewer_JxTL · 2025-06-13

**Clarity:** 2
**Significance:** 2
**Originality:** 2
**Rating:** 4
**Confidence:** 4

**Summary:**

In this paper, the authors present a two-agent framework for learning option trading using deep reinforcement learning. The authors propose using an option position agent responsible for volatility timing and a hedge routing agent to manage risk. The authors compare the proposed approach to 8 different baselines using various evaluation metrics and employ two crypto option datasets (BTC and ETH).

**Questions:**

- The authors claim in l. 47-49 that "options trading is an untouched area due to its complexity," while citing existing papers on the topic (e.g., Tan et al., 2024) without appropriate discussion or comparison. Could the authors clarify why such prior work is not more thoroughly discussed or included in the comparisons, and whether their claims regarding novelty should be revised?

- It is not clear whether commission and transaction costs were incorporated into the reward function or evaluations. Could the authors clarify whether these costs were included, and if so, how they were estimated and implemented?

- The paper does not discuss the stability of the training process or the robustness of the learned policies. Could the authors comment on whether the learned policies are stable and how multiple runs are expected to affect the results?

- In Table 2, almost all baselines (except one) lead to negative profits. Could the authors provide more details on how the baselines were implemented and tuned, to clarify whether these results reflect inherent limitations of the baselines or possible issues in their implementation?

**Ethical Concerns:**

["NO or VERY MINOR ethics concerns only"]

**Final Justification:**

After reading the authors’ rebuttal, I am increasing my score to Borderline Accept, as the authors have addressed most of my concerns. However, I share the view of other reviewers that the methodological novelty (e.g., in the formulation) is limited, and the paper leans more toward the application side. Therefore, I would not oppose rejecting the paper if there is a strong opinion towards this.

**Limitations:**

Yes.

**Paper Formatting Concerns:**

No.

**Quality:**

2

**Strengths And Weaknesses:**

The paper addresses an interesting topic, which has been less studied in recent years despite the wealth of literature on financial trading using DL/DRL. The authors nicely motivate their work and present all necessary background in a concise and easy-to-follow manner. I think the paper holds some value, mainly due to the proposed use of two agents for simultaneously optimizing the objective. However, I also have the following important concerns:

- The authors claim in l. 47-49 that "options trading is an untouched area due to its complexity." However, there are papers on topics very closely related, e.g.:
Tan, Wee Ling, Stephen Roberts, and Stefan Zohren. "Deep Learning for Options Trading: An End-To-End Approach." Proceedings of the 5th ACM International Conference on AI in Finance, 2024.
among others. This paper is also cited by the authors but is not appropriately discussed or compared against. This is a major shortcoming of the work. Therefore, I think the authors should revisit the paper and reconsider some of their strong claims and/or appropriately discuss the existing literature on the topic, which, even though limited, does exist.

- It is not clear whether commission/transaction costs have been incorporated into the reward or evaluations. The authors do mention them, but it is unclear whether they were incorporated and how they were calculated. The impact of different transcation costs on the performance of the method is not quantified.

- The authors do not appropriately discuss the stability of the training process and/or provide evaluations to demonstrate whether the policies learned by the agents are stable (e.g., by conducting multiple training/evaluation experiments and reporting the standard deviation).

- All baselines used in Table 2, apart from one (over 16), result in negative profit. These results raise questions about how well the baselines have been implemented and/or tuned.

- The presentation of the paper can be improved. There are important acronyms defined only in the abstract and not in the main text (IV, RV), acronyms never defined (PnL), formats not defined (e.g., dates in Table 1), etc. Overall, the paper requires very careful proofreading to correct such issues and to ensure that all information necessary for the reader is appropriately provided in the text. The presentation of the main method is also somewhat confusing. The authors provide an overview in Fig. 2, but the presentation order does not follow the logical flow of the method (e.g., the oracle distillation is actually presented at the end).

---

> ### Author Rebuttal · Authors · 2025-07-31
>
> Thank you for your valuable comments. Please find our detailed responses below.
>
> ### **Originality and Related Work Discussion**
> You are correct in pointing out the work by Tan et al. [1]. We acknowledge this important reference and will revise our claims more precisely. While [1] does apply deep learning to options trading, there are key differences in problem formulation and approach:
>
> **Different trading objectives**: [1] focuses on **option portfolio management**, constructing long/short positions across different stocks' straddles to form diversified option portfolios without underlying hedging. Our work specifically targets **volatility trading with dynamic hedging** for single underlying assets, which involves fundamentally different challenges, including gamma scalping, delta hedging, and path-dependent profit optimization.
>
> **Methodological differences**: [1] uses supervised learning for portfolio allocation decisions, while we are the **first to formulate volatility trading as an MDP problem** and apply reinforcement learning to learn optimal volatility timing and hedging strategies jointly.
>
> **Market focus**: [1] addresses equity options portfolio construction, while we focus on cryptocurrency options volatility trading with continuous hedging requirements.
>
> We will revise our manuscript to more rigorously discuss this related work and clarify that our contribution lies in being the first RL framework designed explicitly for **volatility trading with dynamic hedging**, rather than general options trading.
>
> ### **Transaction Costs**
> We do incorporate transaction costs in our simulation:
> - **Commission fees**: Following Deribit's structure - 0.05% for perpetual futures and 0.03% for options (capped at 12.5% of option price)
> - **Bid-ask spread costs**: Implicitly included through realistic market order execution prices in backtesting
>
> Our trading frequency is relatively low (average holding periods of 9-51 hours as shown in Table 3), making transaction costs a small component of total P&L. The following table shows transaction costs as a percentage of total P&L in our experiments:
>
> | Strategy | Options Cost (% of P&L) | Underlying Cost (% of P&L) | Total Cost (% of P&L) |
> |----------|-------------------------|---------------------------|----------------------|
> | OPHR BTC | 4.15% | 5.21% | 9.36% |
> | OPHR ETH | 2.97% | 2.78% | 5.75% |
>
> These results demonstrate that our trading strategy is not sensitive to transaction costs, as the fees represent a relatively small proportion of overall profits. We apologize for not stating this clearly and will add this information to the revised manuscript.
>
> ### **RL Training Stability**
> We acknowledge the importance of training stability analysis. Our framework incorporates several stability-enhancing design elements:
>
> **Oracle initialization**: Phase 1 provides consistent starting points, reducing variance across training runs.
>
> **n-step TD learning**: Effectively reduces noise in Q-value estimation, making RL training convergence more stable than 1-step methods.
>
> **Lower-frequency HR-Agent switching**: The HR-Agent's less frequent hedger selection (every 24 hours) improves training stability by reducing action space complexity and preventing convergence difficulties from frequent switching.
>
> **Relative reward design**: Using the advantage of selected hedgers over baseline hedgers as reward provides more stable learning signals than absolute performance measures.
>
> ### **Baseline Performance and Implementation**
> We provide detailed descriptions of all baseline implementations in Appendix C.3, including consistent principles for position entry/exit, hedging policy, take-profit/stop-loss rules, and holding periods setup.
>
> For each category of baseline strategies, we performed systematic grid search to optimize key hyperparameters as follows:
>
> **Single-Factor Models (Mean Reversion / Momentum)**: Construct percentile-based signals from a 365-day historical RV distribution. Realized volatility is computed over multiple time windows (3–24 hours), and trading is triggered when thresholds on the percentile or its momentum are met.
>
> **Machine Learning Models (GBDT, MLP, LSTM)**: Use predicted annualized RV over a forward window and compare it to current ATM IV to generate trading signals. Entry/exit thresholds were tuned for best validation performance. Fine-tuned take-profit and stop-loss rules were applied.
>
> All baseline strategies were evaluated under the same market environment with performance metrics to ensure fair and consistent comparison.
>
> Despite careful tuning and consistent evaluation, the predominantly negative baseline returns reflect the inherent difficulty of volatility trading rather than implementation issues:
>
> **1. Model assumption burden**: Traditional methods require explicit volatility modeling choices that introduce misspecification risks, while RL learns optimal actions directly without these constraints.
>
> **2. Path-dependent execution gap**: Building on these modeling limitations, optimal hedging decisions depend on complex interactions between market conditions, position Greeks, and future price paths (Section 2.2, Figure 1). ML models provide point forecasts but cannot capture the sequential, path-dependent nature where the sequence of hedging decisions critically impacts P&L. Converting forecasts into profitable trading sequences requires manually designed rules that our grid search experiments showed are difficult to optimize across market regimes.
>
> **Additional baseline details (DLOT)**: While [1] trades multiple assets' straddles without delta hedging (using only straddle P&L as labels), we focus on single-asset volatility trading requiring delta hedging. Therefore, we use delta-hedged straddle P&L as prediction labels. This baseline closely resembles our Phase 1 Oracle policy - one uses supervised learning loss, the other uses offline RL loss.
>
> **Results analysis**: DLOT significantly outperforms RV prediction-based methods, benefiting from an end-to-end design that avoids model assumptions - similar in spirit to our RL approach. However, lacking adaptive hedger selection and further RL optimization, it cannot match our method's performance.
>
> | Method | TR(%)↑ | ASR↑ | ACR↑ | ASoR↑ | AVOL(%)↓ | MDD(%)↓ | WR(%)↑ | PLR↑ |
> |---------|--------|------|------|-------|----------|---------|--------|------|
> | **BTC Delta-hedged P&L** | 4.91 | 0.52 | 0.55 | 0.66 | 21.40 | 8.92 | 47.97 | 1.11 |
> | **ETH Delta-hedged P&L** | 1.19 | 0.07 | 0.09 | 0.09 | 17.14 | 13.53| 42.80 | 1.34 |
>
> This confirms that even sophisticated prediction-based approaches struggle with the execution challenges our RL framework addresses.
>
> ### **Presentation Improvements**
> Thank you for these valuable suggestions. We will comprehensively revise the manuscript to address acronym clarity (defining IV, RV, P&L in the main text), format consistency, logical flow reorganization, and comprehensive proofreading to ensure all technical terms are properly introduced. We appreciate your thorough review and will incorporate these improvements to enhance the paper's clarity and accessibility.
>
> ### **Reference:**
> [1] Deep Learning for Options Trading: An End-To-End Approach. ICAIF 2024.

---

> > ### Comment · Reviewer_JxTL · 2025-08-02
> >
> > Thank you for your responses. Most of my concerns have been addressed by your responses and I am increasing my score.

---

> > > ### Author Response · Authors · 2025-08-02
> > >
> > > Thank you for your thoughtful reconsideration and the increased score. We appreciate your constructive review, which has helped us clarify important aspects of our work and will strengthen the final manuscript.

---

### Official Review · Reviewer_WWUg · 2025-06-24

**Clarity:** 2
**Significance:** 2
**Originality:** 2
**Rating:** 4
**Confidence:** 3

**Summary:**

This paper introduces OPHR, a multi-agent reinforcement learning framework designed for volatility trading in options markets, specifically targeting cryptocurrency derivatives. The primary trading objective is to capture the difference between implied volatility and realized volatility. The framework consists of two specialized agents: the OP-agent, which is responsible for volatility timing by selecting long, short or neutral positions in the parity cross-market. The HR-Agent selects from hedging strategies with different levels of risk aversion to optimize delta hedging under different market conditions. Training is performed in two phases: an offline initialization using an Oracle policy and online iterative joint training of the two agents. The approach is evaluated on BTC and ETH options with superior results over multiple baselines.

**Questions:**

1. Technical details on the Hedger pool construction (e.g., hyperparameter sensitivity of the hedger selection) could be expanded.

2. Considering real-world deployment in live markets, how do you handle the latency of action selection for HR-Agent given that hedging may need near real-time responsiveness, especially during volatility spikes?

3. Have you tested OPHR’s generalizability to other markets (e.g., SPX options, equity index options)?

**Ethical Concerns:**

["NO or VERY MINOR ethics concerns only"]

**Final Justification:**

I have read the rebuttal, and it addressed my main concerns, so I have increased the score.

**Limitations:**

Yes.

**Quality:**

3

**Strengths And Weaknesses:**

### Strengths
1. Thoughtful model design addressing practical market issues like transaction costs and discrete hedging intervals.

2. Novel decomposition into an OP-Agent and a separate HR-Agent. Use of Oracle policy for offline pretraining is well justified.

3. Extensive experiments on real market data and OPHR demonstrates robust returns, reduced drawdowns, improved Sharpe/Calmar/Sortino ratios, and sophisticated volatility timing behaviors.

### weaknesses
1. There are no significant innovations from the perspective of RL methods, and the core contribution lies in the use of existing RL methods for volatility trading.

2. Lack of comparison with more advanced or recent volatility models, such as GARCH variants or deep volatility forecasting networks, which may outperform ML baselines like MLP or LSTM.

3. There is a lack of ablation studies to illustrate the performance improvement and significance of TD learning method design for both OP-Agent and HR-Agent.

4. Limited theoretical analysis. This work has strong application value but lacks profound theoretical basis. However, the code of the results that pay attention to practical application is not open source, which makes it difficult to reproduce and further research, and may reduce its contribution to a certain extent.

---

> ### Author Rebuttal · Authors · 2025-07-31
>
> Thank you for your valuable comments. Please find our detailed responses below.
>
> ### **Innovation from RL Perspective**
> While our framework builds upon established RL techniques, this work represents the **first application of RL to volatility trading in options markets**. Our contributions go far beyond naive application of existing methods - we designed a complete framework specifically for this previously unsolved problem:
>
> 1. **Novel problem formulation**: We introduce the first Cooperative MDP formulation for volatility trading (Section 3), decomposing the complex task into specialized agents with complementary objectives that address the unique interdependence between position sizing and dynamic hedging.
> 2. **Tailored multi-agent training framework**: The two-phase training with Oracle initialization, alternating agent updates, and relative reward design specifically addresses volatility trading's challenges - delayed rewards, noisy signals, and complex state-action dependencies that don't exist in traditional RL applications.
> 3. **Specialized technical details**: Critical implementation details include n-step TD for handling noisy option trading rewards, lower-frequency HR-Agent switching for training stability. These technical choices are essential for making RL work in this domain.
>
> The success of our approach required solving numerous domain-specific challenges from problem formulation through implementation details - simply applying standard RL algorithms would not have succeeded in this complex financial trading environment.
>
> ### **Advanced Volatility Model Baselines**
> Even sophisticated volatility models like GARCH variants or deep volatility networks face the same fundamental limitations we identified:
>
> **1. Model assumption burden**: Advanced volatility models still require explicit modeling choices and assumptions that introduce misspecification risks, while RL learns optimal actions directly without these constraints.
> **2. Forecasting-execution gap**: The core issue isn't prediction quality (Table 10 shows respectable performance) but translating forecasts into profitable sequential trading decisions. Advanced models excel at forecasting but cannot capture the path-dependent execution required for options trading.
>
> **Empirical evidence**: We tested GARCH and DeepVol models:
>
> | Model | TR(%)↑ | ASR↑ | ACR↑ | ASoR↑ | AVOL(%)↓ | MDD(%)↓ | WR(%)↑ | PLR↑ |
> |-------|--------|------|------|-------|----------|---------|--------|------|
> | **BTC GARCH** | -40.83 | -4.29 | -0.95 | -5.54 | 9.51 | 42.76 |26.32 | 0.78 |
> | **BTC DeepVol** | -14.24 | -1.30 | -0.65 | -2.44 | 10.92 | 21.85 | 36.36 | 1.33 |
> | **ETH GARCH** | -52.59 | -3.79 | -0.90 | -4.18 | 13.85 | 58.10 | 12.50 | 0.62 |
> | **ETH DeepVol** | -22.82 | -1.79 | -0.96 | -3.03 | 12.71 | 23.76 | 42.86 | 1.15 |
>
> Despite their sophistication, these models still underperform significantly. The transition from forecasting capability to profitability requires complex model construction and market-specific adjustments - a process our RL method circumvents through direct action optimization without manual model tuning.
>
> ### **Ablation Study on n-step TD Learning**
> We use n-step TD because 1-step TD struggles with convergence due to noisy stepwise reward signals in options trading. The n-step approach is essential for capturing the delayed and cumulative nature of options trading rewards. In volatility trading, profits don't materialize immediately from individual actions but accumulate over multiple time steps through gamma scalping (buying low and selling high as prices oscillate) and theta decay (time value erosion). A single trading decision's profitability depends on subsequent market movements and hedging actions, creating reward signals that are inherently delayed and path-dependent. In our experiments, 1-step TD fails to adequately learn from these profitable experiences because it cannot connect immediate actions to their eventual cumulative rewards, leading to poor strategy development and convergence difficulties. The n-step TD method effectively addresses this by incorporating future rewards over longer horizons, enabling better learning from the temporal structure of options trading profits.
>
> ### **Theoretical Analysis and Reproducibility**
> While we don't provide formal convergence guarantees, our framework incorporates principled design elements that ensure performance stability in practice: Oracle initialization, n-step TD learning for noise reduction, lower-frequency agent switching for training stability, and relative reward design for stable learning signals.
>
> **Regarding reproducibility**: We commit to open-sourcing our complete backtesting environment and RL training framework to enable full reproducibility. We will protect only the specific feature engineering components that provide competitive advantage, ensuring the core methodology remains accessible while balancing academic contribution with practical deployment considerations.
>
> ### **Technical Details on Hedger Pool Construction**
> Comprehensive details are provided in Appendix B.2, including:
> - **Selection criteria**: Top-30 performers based on validation performance across market regimes
> - **Hyperparameter ranges**: Tables 5-6 detail all parameters for Delta-based, Price-based, and Deep Hedgers
> - **Training methodology**: Different risk-aversion levels (λ from 0.1 to 800) ensure diverse hedging behaviors
>
> ### **HR-Agent Responsiveness in Real-time Trading**
> **Training vs. deployment distinction**: During training, the HR-Agent switches hedgers at fixed 24-hour intervals (nhr = 24) to ensure RL training stability - frequent switching would cause training instability and convergence difficulties. However, in live deployment, the HR-Agent can operate in an **event-driven manner**, responding immediately to significant market moves or Greek threshold breaches. The fixed interval is a training constraint, not a deployment limitation.
>
> ### **Generalizability to Different Markets**
> Our focus on cryptocurrency options markets is driven by practical advantages and research value:
>
> **Data accessibility**: Crypto derivatives provide comprehensive 24/7 data access essential for robust RL training, unlike traditional options markets with limited historical data availability.
>
> **Research value**: As an emerging market, crypto options offer higher research value and cleaner microstructure without complex market maker intermediation, allowing focus on core volatility trading dynamics.
>
> **Framework generalizability**: Our RL methodology can extend to different markets, with success primarily depending on market-specific feature engineering. Even within our study, BTC and ETH required different feature engineering approaches to achieve optimal results (though all baselines for each asset used identical features). However, feature engineering optimization falls outside this research's scope - our contribution lies in the RL framework design rather than market-specific feature selection.
>
> Our research aims to demonstrate that RL methods indeed have advantages over traditional approaches in the volatility trading domain.

---

> ### Author Response · Authors · 2025-08-05
> **Additional Details on Advanced Volatility Model Baselines**
>
> **Additional Details on Advanced Volatility Model Baselines**
>
> As mentioned in our earlier response, we implemented and evaluated volatility forecasting baselines using both GARCH and deep volatility forecasting networks for realized volatility prediction. Below, we provide additional details regarding their implementation.
>
> - **GARCH:** We implemented the standard **GARCH** model, trained on rolling daily returns using a one-year window. It was used to predict the realized volatility for the following 24-hour horizon.
> - **DeepVol**: We implemented the **DeepVol** architecture proposed by Moreno-Pino and Zohren (2024) [1], which employs **dilated causal convolutions** to capture long-range dependencies from 5-minute return sequences. The model was trained using a one-year rolling window and used to predict the realized volatility for the following 24-hour period.
>
> To ensure consistency across experiments, we employed the **same decision-making logic** as used in other baselines, conducted hyperparameter tuning for each model, and evaluated both **GARCH** and **DeepVol** using the **same delta-based hedger** in other baselines.
>
> We hope this additional details provides a clearer understanding of our supplementary experimental setup and shows the inherent difficulty in converting volatility forecasts into trading gains. We remain happy to address any further questions you may have.
>
> [1] Fernando Moreno-Pino and Stefan Zohren. *DeepVol: Volatility Forecasting from High-Frequency Data with Dilated Causal Convolutions*

---

### Official Review · Reviewer_vLxe · 2025-07-04

**Clarity:** 3
**Significance:** 3
**Originality:** 2
**Rating:** 4
**Confidence:** 5

**Summary:**

This paper introduces OPHR, a novel multi-agent deep reinforcement learning framework for volatility trading in options markets. The core idea is to decompose the complex trading task into two specialized roles: an **Option Position Agent (OP-Agent)**, which determines the overall strategy (long or short volatility) by timing the market, and a **Hedger Routing Agent (HR-Agent)**, which dynamically selects the optimal hedging policy from a pool of pre-trained hedgers to manage risk and maximize path-dependent profit. The framework is trained in two phases: an offline initialization phase where the OP-Agent learns from a privileged "Oracle" policy, followed by an iterative online phase where both agents are trained collaboratively. The authors evaluate their framework on historical Bitcoin and Ethereum options data from 2021-2024, demonstrating that OPHR significantly outperforms traditional strategies, factor models, and machine learning baselines across a comprehensive set of performance and risk metrics.

**Questions:**

1.  The baseline ML methods (GBDT, MLP, LSTM) are designed to forecast future realized volatility, and Table 10 shows they achieve respectable forecasting performance (e.g., Pearson IC > 0.6, R² > 0.57 for BTC). However, their trading performance in Table 2 is very poor. The authors correctly state that the ML baselines "fail to bridge the gap between forecasting RV and optimizing path-dependent PnL outcomes" (Lines 270-272). Could you elaborate further on why this gap is so large? Is it primarily because a simple forecast-to-trade rule cannot capture the path-dependent effects of hedging and costs, or is it that these models are inherently poor at predicting the rare, fat-tailed volatility events where most of the profit from long-gamma strategies is generated (as suggested in Appendix D.2)? A deeper analysis here would strengthen the paper's core thesis.

2.  Following on from Weakness #2, the reliance on BSM for Greek calculation seems to be a significant model assumption. Have you considered or experimented with more sophisticated methods to obtain more accurate portfolio sensitivities? For example, one could use a model that accounts for the volatility smile (e.g., SABR) or employ model-free Greek estimates. How would the performance of the HR-Agent, and thus the entire OPHR framework, be affected if it were provided with more accurate, market-implied Greeks?

3.  The Oracle policy used for offline pre-training (Phase 1, Line 210) leverages future realized volatility to generate "desirable and profitable trading behaviours". This is a clever way to bootstrap the agent. Could you provide some insight into the sensitivity of the final OPHR performance to the quality of this pre-training? For instance, how does the performance change if the Oracle's `β` threshold is defined differently or if the pre-training phase is shortened or skipped entirely?

**Ethical Concerns:**

["NO or VERY MINOR ethics concerns only"]

**Final Justification:**

The concerns were addressed by author's rebuttal.

**Limitations:**

Yes

**Paper Formatting Concerns:**

No formatting issues.

**Quality:**

2

**Strengths And Weaknesses:**

### Strengths

1.  **Novel and Sensible Problem Decomposition:** The paper's primary strength lies in its intelligent decomposition of the volatility trading problem into a multi-agent system. Separating the strategic decision of "volatility timing" (OP-Agent) from the tactical execution of "risk management" (HR-Agent) is a clear and logical approach that mirrors how human traders might operate. This hierarchical structure (explicitly formulated as a Cooperative MDP in Section 3, page 4) is well-suited for the complexities of options trading and represents a significant and original contribution to the application of RL in finance.

2.  **Thorough and Robust Empirical Evaluation:** The experimental setup is comprehensive and of high quality. The authors test their model on a multi-year dataset (Table 1) covering diverse market regimes (bull, bear, high volatility). They compare OPHR against a strong and varied set of baselines, including directional strategies, single-factor models, and standard ML forecasting models (Section 5.1). The results, particularly in Table 2, show a consistent and substantial outperformance of OPHR, not just in total returns but, more importantly, in risk-adjusted metrics like Sharpe, Calmar, and Sortino ratios. The ablation study (comparing `OPHR` to `OP`) effectively demonstrates the value added by the adaptive HR-Agent.

3.  **High-Quality Presentation and Reproducibility:** The paper is well-written and clearly structured. Figures 1 and 2 provide excellent visual intuition for the core challenges (hedging) and the proposed architecture. Furthermore, the appendices are exceptionally detailed, providing full transparency on the experimental setup. The inclusion of detailed hyperparameter tables (Tables 4, 5, 6), baseline implementations (Appendix C.3), and evaluation metric definitions (Appendix C.1) significantly strengthens the paper and provides a clear path for reproducing the work.

### Weakness

1.  **Omission of Transaction Costs:** The authors correctly state that "frequent hedging incurs transaction costs" (Line 181), identifying this as a key real-world challenge. However, the simulation environment appears to be frictionless, omitting trading fees, bid-ask spreads, and potential slippage. This is a critical limitation. A core component of the OPHR strategy is dynamic hedging, which can involve frequent trades. In real markets, these transaction costs can accumulate rapidly and significantly erode, if not completely eliminate, the reported profits. The omission undermines the practical viability of the strategy and weakens the claim of outperformance.

2.  **Unrealistic Assumption for Greek Calculation:** The framework's state representation and risk management rely heavily on option Greeks (Δ, Γ, Θ, Vega), as stated in Line 134 and used by the HR-Agent. The paper details the standard Black-Scholes-Merton (BSM) formulas for these Greeks in Appendix A.1.3. Using BSM, which assumes a single, constant volatility, is highly unrealistic for cryptocurrency options markets that exhibit pronounced volatility smiles and skews. This means the Greeks used as input to the HR-Agent are likely inaccurate. The very high "Residual" PnL reported in the PnL attribution analysis (Table 13, e.g., -51.45% for BTC Long and 153.03% for ETH Short) serves as evidence that the BSM model fails to explain a large portion of the portfolio's returns, directly highlighting the model's inadequacy in this context.

---

> ### Author Rebuttal · Authors · 2025-07-31
>
> Thank you for your thorough and insightful review. Please find our responses below.
>
> ### **Omission of Transaction Costs**
> We apologize for not clearly stating this in the paper. We **do** consider transaction costs in our simulation, which are deducted from the cash balance when trades are executed. Our implementation follows Deribit's fee structure: **Commission fees**: 0.05% for perpetual futures and 0.03% for options (capped at 12.5% of option price). We also consider the **Bid-ask spread costs**: We use market orders in backtesting, so spread costs are implicitly included through realistic execution prices.
>
> Importantly, our trading policy is **not high-frequency**. As shown in Table 3, average holding periods range from 9-21 hours for long positions and 47-51 hours for short positions. The HR-Agent selects hedgers every 24 hours (Table 6), making transaction costs a relatively small component of total P&L.
>
> The following table shows transaction costs as a percentage of total P&L in our experiments:
>
> | Strategy | Options Cost (% of P&L) | Underlying Cost (% of P&L) | Total Cost (% of P&L) |
> |----------|-------------------------|---------------------------|----------------------|
> | OPHR BTC | 4.15% | 5.21% | 9.36% |
> | OPHR ETH | 2.97% | 2.78% | 5.75% |
>
> We will add this crucial information to the revised manuscript.
>
> ### **Unrealistic Assumption for Greek Calculation & Advanced Vol Smile Models**
> We acknowledge that BSM makes simplifying assumptions, but our framework is designed to be robust to Greek calculation imprecision for several key reasons:
>
> **1. Greeks as reference values, not precise inputs**: Our method doesn't rely on perfect Greek precision but uses them as reference signals. BSM remains widely used by market participants despite its limitations - experienced traders understand these constraints and adapt accordingly.
>
> **2. Data-driven error correction**: The Deep Hedgers take Greeks as input features, and the HR-Agent routes among hedgers based on their actual performance during RL training. This means model misspecification is automatically "priced in" through the data-driven learning process, similar to how human traders compensate for known BSM limitations.
>
> **3. Residual PnL explanation**: The high residuals in Table 13 are primarily due to discrete time intervals (5-minute data) rather than Greek misspecification. Second-level data would yield much smaller residuals, as the Taylor expansion in Eq. (2) assumes continuous hedging. Our PnL decomposition serves to qualitatively analyze whether our strategy's profit sources align with expectations (short positions earning theta, long positions earning gamma).
>
> **Why advanced smile models aren't necessary**: More sophisticated volatility smile models (e.g., SABR) focus on precise pricing across the entire volatility surface, typically used for market-making or volatility surface arbitrage. Our strategy trades only ATM straddles with relatively low frequency, making precise smile modeling unnecessary for our specific application.
>
> ### **Why ML-based Strategies Fail**
> The poor performance of ML baselines despite good forecasting metrics (Table 10 vs Table 2) highlights fundamental challenges in translating predictions to profitable trading:
>
> **1. Model assumption burden**: ML forecasting requires explicit volatility modeling choices (estimators, frequencies, etc.) that introduce model misspecification risks. RL circumvents these assumptions through direct action optimization, adapting to market dynamics that traditional volatility frameworks may not capture.
>
> **2. Forecasting-execution gap**: Building on these modeling limitations, ML models excel at volatility forecasting but struggle with the sequential, path-dependent execution required for options trading. As shown in Section 2.2 and Figure 1, optimal hedging decisions depend on complex interactions between market conditions, position Greeks, and future price paths. Converting point forecasts into profitable trading sequences requires manually designed rules that are difficult to optimize across market regimes - precisely the challenge our grid search experiments revealed.
>
> These fundamental issues (1 & 2) affect all prediction-based approaches, while ML methods also face an inherent limitation:
>
> **3. Fat-tail prediction failure**: While ML models achieve good overall forecasting performance, Table 11 shows they perform poorly on extreme volatility events (top 5% outliers) where most long-gamma profits are generated. RL's exploration and adaptation capabilities provide advantages precisely in these critical profit-generating scenarios.
>
> **End-to-end ML forecasting baseline**: We add an end-to-end ML baseline that circumvents problems 1 & 2 following [1]. Please refer to the detailed experiment results in section **Additional baseline details (DLOT)** in our response to **Reviewer JxTL**. Results show that supervised learning methods still have significant disadvantages compared to RL in this scenario, demonstrating the superiority of RL's adaptive learning approach even when modeling assumptions are minimized.
>
> ### **Oracle Policy Sensitivity Analysis**
>
> The Oracle policy serves as a crucial bootstrap mechanism during Phase 1. To ensure diverse and profitable experience generation, we adopt a range of β thresholds when constructing the Oracle policy. Specifically, as shown in Table 4, we vary the Oracle threshold β across multiple values (0.1, 0.2, 0.4, 0.6, 0.8) to produce a broad spectrum of trading signals. This diversity encourages the OP-Agent to learn a richer set of profitable behaviors during pre-training.
>
> These varied Oracle signals are then incorporated and refined in Phase 2, where the agent learns to adaptively combine and build upon these strategies. This two-phase approach is critical due to the delayed nature of profit realization in volatility trading, providing structured guidance toward profitability early in training.
>
> Based on our design rationale, the Oracle need not be optimal - it simply needs to demonstrate profitable trading patterns that reduce exploration burden. Shortening or skipping the pre-training phase would lead to significantly slower convergence, as the agent would lack initial guidance and struggle to discover effective volatility trading strategies through exploration alone amid market noise. We will incorporate these explanations into the revised manuscript for clarity.
>
> ### **Reference:**
> [1] Deep Learning for Options Trading: An End-To-End Approach. ICAIF 2024.

---

### Official Review · Reviewer_KnBe · 2025-07-19

**Clarity:** 3
**Significance:** 4
**Originality:** 3
**Rating:** 5
**Confidence:** 4

**Summary:**

In this paper, the authors study the volatility trading task in the option markets. A multi-agent framework is proposed, which consists of two types of agents, OP-agent and HR-agent, respectively. Crypto option data was used in the experiments for verifying the effectiveness.

**Questions:**

Plz provide justifications to the above questions.

As mentioned before, this reviewer does not have solid expertise to evaluate the reported results. One extra question for clarification is: should the TWAP and VWAP be considered as baseline methods for this work? If yes, plz add those results as baselines too.

**Ethical Concerns:**

["NO or VERY MINOR ethics concerns only"]

**Final Justification:**

Thanks for the authors' responses. They address my concerns well. And I will keep my scores.

**Limitations:**

The reviewer is not aware of such concerns.

**Paper Formatting Concerns:**

It does not apply here.

**Quality:**

3

**Strengths And Weaknesses:**

The strengths can be summarized as follows:
1. The volatility trading task in the option markets is fundamentally important. It is a critical task in the finance sector. The reviewer believes that it could potentially be very useful in the real-world.
2. It is sound to recast the task as a mdp process and then taking an RL approach. The authors provide rigorous formulation in Section 3.
3. The two-phase multi-agent framework is sound. It is very interesting to see an agent pool for a real-world scenario.
4. The experiment results based on real-world data is appreciated. However, this reviewer does not have solid expertise to evaluate the reported results.

However, this paper cannot be fully appreciated due to the following weaknesses:
1. It is NOT clearly presented why RL is the right solution for such a real-world task.  The reviewer is interested to see such an RL solution, however, it is also important to understand why it is the right one?  Is RL a unique choice here?
2. What is an intuitive understanding of the two-phase approach?  Plz help the reviewer and readers to understand the design from a high-level.
3. Is the Deep Hedger reproduced in the experiments?
4. Is it possible to provide any performance guarantee?  Or it can only take an engineering approach?
    many RL research at NeurIPS has deep analysis.  It would be good to show the depth of this work.
    Is there any unique property for RL algorithms to deal with option trading data or crypto data?

---

> ### Author Rebuttal · Authors · 2025-07-31
>
> Thank you for your valuable comments and positive assessment. Please find our detailed responses below.
>
> ### **Why RL is the right solution**
> Traditional approaches that rely on explicit volatility forecasting followed by rule-based trading decisions face fundamental limitations that make RL particularly well-suited for volatility trading:
>
> 1. **Minimizing model assumptions**: Traditional methods depend heavily on explicit modeling assumptions - IV calculations rely on option pricing models such as BSM, while RV estimation requires choosing among different variance estimators (maximum likelihood, maximum posterior, different data frequencies, etc.). In practice, selecting the appropriate models and parameters across varying market conditions is extremely challenging and often leads to model misspecification. RL circumvents these issues through data-driven learning that incorporates all market dynamics directly into the reward signal, eliminating the need for multiple separate models to estimate volatility components.
> 2. **Sequential decision-making & Path-dependent profit**: Volatility trading presents inherently sequential challenges where current decisions affect future risk exposures and hedging costs/timing - this is not a static optimization problem. As shown in Section 2.2 and Figure 1, optimal hedging decisions depend on complex interactions between market conditions, position Greeks, and future price paths. Current option positions determine future delta exposure, which influences subsequent hedging requirements and costs. In our experiments, we conducted extensive grid search optimization for rule-based strategies but still struggled to find configurations that generate stable profits across different market regimes. Traditional approaches using predictions with manually designed rules are prone to getting trapped in local optima or failing entirely. RL naturally handles these sequential dependencies by learning optimal action sequences that account for future implications of current decisions.
>
> ### **Intuition of the two-phase approach**
> Oracle policy initialization has been used in trading RL [1,2], but volatility trading presents unique challenges that motivate our two-phase design. The reward structure is highly path-dependent, and profitable trading trajectories are relatively sparse - making it extremely difficult to discover profitable strategies through random exploration alone. Moreover, the few profitable trajectories can be easily overwhelmed by noisy market data, leading to poor convergence.
>
> Our Oracle policy uses delta-hedged P&L signals to guide the agent toward profitable trading behaviors, significantly reducing exploration costs. This initialization ensures the agent starts from a reasonable region of the policy space, then RL training further optimizes beyond this foundation to discover superior strategies that pure exploration would struggle to find amid market noise.
>
> ### **Deep Hedger reproduction**
> Yes, we reproduce the Deep Hedger following [3]. Appendix B.2 describes the details of different hedgers including Deep Hedgers, and Table 5 shows the parameters used for these hedgers. Our implementation follows the actor-critic framework with risk-adjusted utility functions and varying risk-aversion parameters as described in the original work.
>
> ### **Performance Guarantee**
> **Theoretical considerations**: While we don't provide formal convergence guarantees as this is an empirical RL research, in our research practice we found the following elements crucial for performance assurance:
>
> 1. **Oracle policy initialization**: The Oracle policy in Phase 1 ensures we start from a profitable region of the policy space, reducing the risk of converging to poor local optima.
> 2. **n-step TD learning**: Effectively reduces noise in Q-value estimation, making RL training convergence more stable compared to 1-step methods.
> 3. **Lower-frequency HR-Agent switching**: The HR-Agent's less frequent hedger selection (every 24 hours) similarly improves training stability by reducing action space complexity.
> 4. **Relative reward design for HR-Agent**: Using the advantage of selected hedgers over baseline hedgers as reward provides more stable learning signals than absolute performance measures.
>
> ### **TWAP/VWAP baselines**
> These execution algorithms are designed for minimizing market impact when trading large orders, which is orthogonal to our volatility trading objective. Our baselines already comprehensively cover the relevant comparison space: directional strategies (Long/Short), factor models (MR/MOM), and ML forecasting approaches (GBDT/MLP/LSTM). TWAP/VWAP would not provide meaningful comparisons for volatility timing and Greek management, as they address fundamentally different trading problems.
>
> ### **References:**
> [1] Universal Trading for Order Execution with Oracle Policy Distillation, AAAI 2021
> [2] EarnHFT: Efficient Hierarchical Reinforcement Learning for High Frequency Trading, AAAI 2024
> [3] Deep Hedging: Continuous Reinforce359 ment Learning for Hedging of General Portfolios across Multiple Risk Aversions, ICAIF, 2022

---

> ### Comment · Reviewer_KnBe · 2025-08-08
>
> I put in the justification that "Thanks for the authors' responses. They address my concerns well. And I will keep my scores."
>
> One quick question: Are the datasets and implementation codes publicly available?   If not, will them be released in the future?
>
> There is no more concern from my end. The rest is left to the meta-reviewer and AC. Good luck!

---

> > ### Author Response · Authors · 2025-08-09
> >
> > Thank you for your positive feedback and support throughout the review process.
> >
> > Regarding code and data availability: We promise to open-source the backtest environment and RL code in the camera-ready version. Due to our data provider's terms, we cannot distribute the raw datasets. To facilitate reproduction of our work, we will provide sample data with corresponding sample scripts. Here we briefly describe our feature engineering pipeline: We obtain tick-level perpetual futures and options data from Tardis. The perpetual data is downsampled to construct technical indicators as base features. We then perform time-rolling feature engineering using multi-horizon returns and realized volatility as targets, applying temporal standardization. The sample data and scripts will demonstrate our complete methodology, enabling researchers to replicate our approach with their own datasets.
> >
> > Thank you again for your thorough review.

---

### Note · Authors · 2025-08-15

We thank all reviewers for their constructive feedback. Here's how we addressed the main concerns:

**Transaction Costs (Reviewers vLxe, JxTL)**: We clarified that transaction costs are included following Deribit's fee structure. Our low-frequency trading makes fees represent only 5.75-9.36% of total P&L, demonstrating strategy robustness.

**Greek Calculation Assumptions (Reviewer vLxe)**: We explained our framework's robustness to BSM limitations through: (1) using Greeks as reference signals rather than precise inputs, (2) data-driven error correction via RL training that automatically accounts for model misspecification.

**RL Innovation & Baselines (Reviewers WWUg, JxTL)**: We emphasized this is the first RL application to volatility trading, requiring novel problem formulation, specialized multi-agent architecture, and domain-specific techniques adaptations (n-step TD, Oracle initialization, relative rewards). We added advanced ML prediction baselines (GARCH, DeepVol), showing that even sophisticated forecasting models fail due to execution gaps that the end-to-end RL method addresses. The end-to-end DL baseline (DLOT) shows our RL method.

**Originality vs. DLOT. (Reviewer JxTL)**: We clarified the distinction - their work focuses on portfolio management without hedging, while ours targets single-asset volatility trading with dynamic hedging, representing fundamentally different problem formulations.

**Training Stability (Reviewer WWUg)**: We detailed stability-enhancing elements: Oracle initialization, n-step TD for noise reduction, lower-frequency HR-Agent switching, and relative reward design.

**Reproducibility (Reviewer WWUg)**: All our experiments use identical data processing pipelines. We promise to open-source the backtesting environment and RL framework. Due to the data provider's terms, we cannot distribute raw datasets. We will provide sample data with execution scripts demonstrating our complete methodology from tick-level Tardis data to technical indicators and multi-horizon features. This enables researchers to replicate our approach using their feature engineering with our open-sourced framework.

Our responses have addressed all the main concerns of the reviewer, and we hope the final remarks can help the AC's consideration.

---

### Decision · Program_Chairs · 2025-09-17

**Decision:**

Accept (poster)

**Comment:**

The paper proposes a two-agent RL framework for options volatility trading—an OP-Agent times long/short/neutral exposure while an HR-Agent routes among hedgers to manage delta and path-dependent P&L—formulated as a cooperative MDP and trained via oracle-initialized offline pretraining plus joint online learning. Reviewers praise the decomposition, careful engineering (n-step TD, lower-frequency HR switching, relative rewards), and broad empirical gains on BTC/ETH options, especially after the rebuttal added advanced volatility baselines (GARCH, DeepVol, DLOT) and clarified transaction-cost modeling (fees amounting to ~6–9% of P&L). Remaining concerns center on modest RL novelty, reliance on BSM Greeks, lack of formal guarantees, limited multi-seed/stability reporting, and generalization beyond crypto.